# A Survey of Data Augmentation Techniques for Traffic Visual Elements

**DOI:** 10.3390/s25216672

**Published:** 2025-11-01

**Authors:** Mengmeng Yang, Lay Sheng Ewe, Weng Kean Yew, Sanxing Deng, Sieh Kiong Tiong

**Affiliations:** 1Institute of Sustainable Energy (ISE), College of Engineering, Universiti Tenaga Nasional, Kajang 43000, Malaysia; laysheng@uniten.edu.my (L.S.E.); dsx3476@163.com (S.D.); siehkiong@uniten.edu.my (S.K.T.); 2School of Mechanical and Electrical Engineering, Huanghe Jiaotong University, Jiaozuo 454000, China; 3School of Engineering and Physical Sciences, Heriot-Watt University Malaysia, Putrajaya 62200, Malaysia

**Keywords:** data augmentation, traffic visual elements, GAN, diffusion models, innovation strategy, evaluation metrics

## Abstract

**Highlights:**

**What are the main finding?**

**What are the implications of the main findings?**

**Abstract:**

Autonomous driving is a cornerstone of intelligent transportation systems, where visual elements such as traffic signs, lights, and pedestrians are critical for safety and decision-making. Yet, existing datasets often lack diversity, underrepresent rare scenarios, and suffer from class imbalance, which limits the robustness of object detection models. While earlier reviews have examined general image enhancement, a systematic analysis of dataset augmentation for traffic visual elements remains lacking. This paper presents a comprehensive investigation of enhancement techniques tailored for transportation datasets. It pursues three objectives: establishing a classification framework for autonomous driving scenarios, assessing performance gains from augmentation methods on tasks such as detection and classification, and providing practical insights to guide dataset improvement in both research and industry. Four principal approaches are analyzed, including image transformation, GAN-based generation, diffusion models, and composite methods, with discussion of their strengths, limitations, and emerging strategies. Nearly 40 traffic-related datasets and 10 evaluation metrics are reviewed to support benchmarking. Results show that augmentation improves robustness under challenging conditions, with hybrid methods often yielding the best outcomes. Nonetheless, key challenges remain, including computational costs, unstable GAN training, and limited rare scene data. Future work should prioritize lightweight models, richer semantic context, specialized datasets, and scalable, efficient strategies.

## 1. Introduction

With the rapid pace of modernization and the widespread implementation of Automatic Driving Technology, the demand for accuracy and real-time object detection in Intelligent Transportation Systems has increased. Currently, the most commonly used methods rely on Deep Learning-based model architectures to ensure the accuracy and efficiency of object detection. However, the performance of these models heavily relies on the quality and diversity of the training data [1,2]. Unfortunately, existing traffic scene datasets are often difficult to collect and lack diversity, extreme scene samples, and balanced category distribution. These issues significantly impact the model’s performance and practical application [3]. Therefore, it is crucial to explore ways to optimize data quality and improve the robustness and detection performance of the model through data augmentation technology, which has become a critical research direction in the field of Intelligent Transportation. Data augmentation technology utilizes specific transformation or expansion methods to generate new data from the original dataset, thereby enhancing the richness and sufficiency of training data and improving the model’s generalization ability and robustness. This technique is widely used in various fields, including computer vision, natural language processing, and speech recognition [4,5].

On the other hand, while the Vehicle’s Automatic Driving System is in operation, it relies on visual perception to understand the surrounding environment in real time and make safety decisions. Traffic visual elements, such as traffic signs [6,7,8], traffic lights [9,10,11], and pedestrians [12,13,14], are crucial sources of information for the system to perceive the environment. Traffic signs and signal lights are crucial components of road safety, but they are often small and vulnerable to environmental interference. These physical markers serve as the foundation for road rules and play a significant role in ensuring safe driving behavior. However, pedestrians and vehicles are constantly moving and can take on various forms, making them difficult to detect and avoid. As such, it is the ethical responsibility of autonomous driving to prioritize the protection of personal safety and prevent traffic accidents. Unfortunately, factors such as wear and tear of road markings, obstructions, and diverse structures can create challenges for autonomous vehicles, hindering their ability to perceive and make decisions effectively and resulting in a lack of robustness. These elements not only assist vehicles in identifying road rules but also directly impact their ability to make timely and accurate driving decisions [15,16]. Therefore, it is crucial to study methods for enhancing the data of traffic visual elements [17,18,19].

Currently, there are limited published reviews on data augmentation techniques for traffic visual elements. This paper aims to address this gap by systematically summarizing the data augmentation methods specifically for traffic visual elements in the field of Automatic Driving. This is crucial in the training stage of object detection models. The structure of this paper is as follows: the second part will discuss relevant augmentation methods, challenges, and innovative strategies, and analyze their application in different traffic visual elements. The third part will introduce relevant datasets and corresponding evaluation indicators for traffic visual elements. The fourth part will highlight future research directions for data augmentation of traffic visual elements. Finally, the fifth part will provide a summary of the research findings. The goal of this paper is to thoroughly examine data augmentation techniques for traffic visual elements in order to provide high-quality training data for the Automatic Driving System. This will ultimately improve the accuracy and robustness of the object detection model and further advance the development of Intelligent Transportation Systems.

Unlike earlier works that primarily examined general image augmentation or single-object datasets, this paper focuses on data augmentation for traffic visual elements, a field where dataset diversity, rare scene representation, and real-world relevance remain insufficiently explored. This study establishes a unified taxonomy that connects augmentation techniques such as image transformation, GAN-based, diffusion, and composite methods to their performance across nearly 40 benchmark datasets.

## 2. Method Overview

The literature search was conducted in February 2025. Databases searched included Scopus, IEEE Xplore, ScienceDirect, and Google Scholar. Web of Science and ACM Digital Library were also considered, but due to significant overlap with Scopus and IEEE Xplore, they were not included separately. The initial search terms comprised “data augmentation,” “traffic scene,” “traffic sign,” “traffic light,” “pedestrian,” “car,” “generative adversarial network,” “diffusion model,” with refined Boolean search strings applied to improve accuracy and coverage. Only peer-reviewed journal and conference papers published in English between 2020 and 2025 were included. Studies outside the transportation domain, non-peer-reviewed sources (e.g., theses, blogs), duplicate records, and papers lacking methodological details were excluded. The screening process involved two stages: (1) titles and abstracts were reviewed for relevance, and (2) full texts were evaluated against the inclusion and exclusion criteria. To minimize omissions, reference tracking (snowball method) was used to identify additional studies. All selected papers were cross-checked by two authors to ensure consistency. A PRISMA flow diagram (Figure 1) is provided to illustrate the paper selection process, and special attention was given to studies involving widely used benchmark datasets to ensure comprehensive coverage.

## 3. Materials and Methods

With the rapid development of automatic driving, computer vision technology has become increasingly prevalent in traffic scenes. This includes tasks such as traffic light detection, traffic sign classification, and pedestrian recognition. However, traffic scenes are complex and constantly changing, making it challenging for models to accurately detect and classify objects. Factors such as adverse weather conditions, varying light intensity, and obstructions from both artificial and natural sources pose significant challenges to the robustness and generalization ability of these models. Additionally, these factors make it difficult to obtain sufficient training data. To address these challenges, data augmentation technology has emerged as a valuable tool. Not only does it enrich the training data, but it also improves the performance of the model and its effectiveness in real-world scenarios. This is crucial for ensuring the reliability and safety of Intelligent Transportation Systems. In this paper, we will review the selection of data augmentation techniques, application strategies, experimental settings, and performance evaluation. Our goal is to provide efficient data processing solutions for traffic vision tasks and offer theoretical guidance and practical advice for researchers in related fields.

### 3.1. Image Transformation Data Augmentation

In the application of data augmentation for traffic visual elements, the primary method of image processing is image transformation. This technique is divided into two categories: simple and comprehensive, as shown in Table 1 and Table 2, respectively. Image transformation data augmentation simulates real traffic scenarios, such as changes in lighting, weather, and occlusion, by altering the geometry, color, or structure of the original image. This increases the diversity of training data and helps to address the issue of low recognition rates caused by blocked traffic signs in the process of Automatic Driving. In a study conducted by ANDREW DINELEY et al., random occlusion with varying degrees of obstruction was used to enhance the data of traffic sign images. The models used for training included AlexNet, VGG19, ResNet50, and GoogLeNet. The results showed that when using GoogLeNet, the recognition accuracy of this augmentation method improved by 17% under high occlusion percentages of 61–70%, and slightly improved under low occlusion percentages. This demonstrates that the proposed data augmentation technology can significantly enhance the recognition performance of the model, even when the traffic sign image is heavily occluded [20]. To significantly improve the detection performance of traffic signs in complex environments and effectively extract the image features, Jingyi Shi et al. proposed a FlexibleCP (Flexible Cut and Paste) data augmentation strategy based on a mosaic. This strategy generates a larger and more diverse training dataset by utilizing techniques such as clipping, copying, transformation, filtering, and pasting. As a result, the model’s robustness and training speed are improved. When trained on the CTSD dataset, the FlexibleCP augmentation strategy showed a 3.5% improvement in mAP0.5 and a 2.5% improvement in mAP0.5:0.95 compared to the mosaic augmentation method [21]. The detection and recognition of traffic lights is a common area of research in traffic scene data augmentation. Huei-Yung Lin and their team proposed a new framework for directly detecting and classifying single traffic lights. This aims to overcome the limitations of relying too heavily on identifying traffic light boxes, as well as the challenges posed by the small size and variable colors of traffic lights. The proposed framework involves two main steps: first, the use of color transformation augmentation technology to synthesize images, and second, the adoption of an integrated learning framework training model combined with recursive data augmentation to improve performance. The results show a recognition accuracy of 97.26% and a classification accuracy of 98.6% (Lin and Chen, 2024) [17].

Basic image processing has a low computational cost. Simple rotation, translation, and other operations can increase the model’s adaptability to different sizes, shapes, and other objects. However, the relevance of the operation task is insufficient, and it is difficult to simulate a complex real scene, so it is necessary to combine more advanced augmentation technology to improve the performance of the model. Naifan Li et al. explored the data augmentation of traffic scenes such as traffic cones, traffic barrels, and triangle warning signs, which are rare in Automatic Driving. The author utilized image transformation methods, such as color changes and scaling, to modify the object masks in the source domain. Additionally, the global context information of traffic scenes, such as roads and lanes, was used to guide the placement of object instance masks. This ensured global consistency within the traffic scene while also allowing for local adaptation of the instance mask to be pasted onto the object image, resulting in the synthesis of rare traffic object training data [22]. Furthermore, in addition to dealing with rare data, class imbalance poses a significant challenge in traffic scene data augmentation. In an effort to improve road safety, Ulan Alsiyeu et al. proposed a novel augmentation technique that incorporates geometric transformations, image synthesis, and obstacle data augmentation (such as superimposing trees and pedestrians onto traffic signs) to expand the traffic sign recognition dataset. This approach has proven to be highly effective and applicable. The authors report that the YOLOv8 model trained on their customized data augmentation dataset achieved an accuracy of 89.2%, which is 5.5% higher than the model trained on the GTSRB training dataset [23].

### 3.2. Data Augmentation Based on GAN

Because the traditional image transformation data augmentation methods rely on the original data, the changes are limited and cannot generate new features, while the real traffic scenes are complex and highly variable. GAN, which can learn from a small amount of data and generate realistic samples, has been widely used in the data augmentation methods of traffic scenes. Umair Jilani et al. utilized GAN to create composite images of traffic congestion and further improved them through data augmentation and image modification techniques. This allows traffic managers to accurately identify and distinguish traffic congestion. The effectiveness of the proposed 5-layer convolutional neural network model was also confirmed through testing on the enhanced dataset, achieving an impressive accuracy rate of 98.63% [24]. To enhance the adaptability of the autonomous driving model to various road conditions, Ning Chen et al. utilized WGAN-GP and integrated it with traditional image transformation techniques to augment the road texture data. This effectively addressed the issue of limited road texture data [25]. In addition to enhancing the dataset and model generalization ability, we should also pay attention to the performance of object detection. Eman T. Hassan et al. emphasized the importance of semantic consistency augmentation to improve the detection accuracy of traffic lights and used GAN to understand the semantics of the scene by generating a heat map, predict the location of traffic lights, so as inserting traffic lights to achieve the purpose of data augmentation. Through experiments, the GAN model can generate many reasonable and effective traffic signal positions so as to reduce false positives and improve the accuracy of the detection model [26]. In addition, low visibility traffic scenes pose a greater challenge for Automatic Driving. To enhance the ability to perceive traffic in these situations, Kong Li and his team developed a pixel augmentation model, PE-Pix2Pix, based on Pix2Pix for controllable data augmentation. The model incorporates an adaptive alpha channel to achieve a more natural fusion between the generated image and the original image. Additionally, it allows for the adjustment of the visual augmentation effect to cater to different environments [18].

#### 3.2.1. Related Work

In 2014, Goodfellow et al. proposed the Generative Adversarial Network (GAN) [27]. Compared to other generation models, GAN has a stronger generative ability and better performance. By training the generator and discriminator in a confrontational manner, GAN is able to generate high-quality and diverse synthetic data, resulting in remarkable results in unsupervised learning tasks such as image, video, and text generation. The main concept of GAN is to train two neural networks, the generator and the discriminator, through confrontation. These two networks compete against each other: the generator creates data from random noise and gradually approximates the distribution of real data, while the discriminator attempts to accurately distinguish between real and generated data. Throughout the training process, both networks are continuously adjusted and optimized with the ultimate goal of reaching a balanced state, known as the Nash equilibrium. This is achieved when the data distribution generated by the generator is indistinguishable from the real data distribution, rendering the discriminator unable to differentiate between the two. The structure is depicted in Figure 2.

Goodfellow’s paper visually demonstrates the iterative process of GAN, as depicted in Figure 3 [27]. In Figure 3, the black dotted line represents the distribution of real data, the green curve represents the distribution of generated data, and the blue dotted line represents the decision boundary of the discriminator. Initially, the discriminator is able to distinguish between true and false data. Additionally, a certain amount of white noise is added to the input of the discriminator to create a more realistic simulated environment. The noise is represented by *z*, and the process of mapping the noise to the data distribution through the generator is represented by *z* to *x*. The goal of the generative adversarial network is to gradually make the green curve approach the black dotted line, ultimately achieving a consistent distribution of true and false data.

In Figure 3, the process of alternating optimization training for the generative adversarial network is illustrated in four stages (a–d). In the initial state (a), there is a significant disparity between the generated data distribution and the real data distribution. Although the discriminator is able to preliminarily distinguish between true and false data, its discrimination ability is still lacking. State (b) shows the stage of fixed generator and training discriminator. It optimizes the decision boundary of the discriminator, gradually improves the discrimination ability, and makes it easier to distinguish the distribution of true and false data. When the discriminator is gradually improved, it should then be fixed and the generator should be optimized, as shown in state (c). The generator uses the gradient information provided by the discriminator to adjust its parameters so that the generated data distribution converges to the real data distribution. After several iterations, the generator and the discriminator will enter the final state (d), and the discriminator will be unable to distinguish whether the sample is real or generated by the generator. The generator can complete the mission of generating realistic images.

In the above training process, the loss function of the generative adversarial network is:(1)$$\underset{G}{min}\;\underset{D}{max}\;V(D,G)=E_{xP_{\mathrm{data}\;}(x)}[logD(x)]+E_{zP_{z}(z)}[log(1-D(G(z)))]$$
where $P_{\mathrm{data}\;}(x)$: real data distribution; $P_{z}(z)$: noise distribution; D(x): The discriminator output probability for real data *x* is taken as [0,1], where 1 represents the real sample, and 0 represents the fake sample. D(G(z)): The discriminator output probability for generated data. In the training process, if D(G(z)) is closer to 1, the expected loss $E_{xP_{\mathrm{data}\;}(x)}[logD(x)]$ of the real data is close to the maximum, and the discriminator can distinguish the real samples as much as possible. If D(G(z)) is closer to 1 and 1−D(G(z)) is closer to 0, the expected loss $E_{zP_{z}(z)}[log(1-D(G(z)))]$ of the generated data is close to the minimum, and the generator can generate real samples as much as possible. The generator and discriminator take turns playing the game repeatedly in order to achieve the most effective generative adversarial network [28].

Thanks to the outstanding performance of GAN, numerous improved versions have been developed for traffic visual scene data augmentation. Dongjun Zhu et al. proposed the first data augmentation framework, MCGAN, to enhance optical remote sensing image object detection. The architecture consists of a generator, three discriminators, and a classifier. Its special feature is that the multi-branch extended convolution is designed in the discriminant network to extract the global, local, and detailed features of the object, thus helping the generator produce more diversified and higher-quality remote sensing images, such as vehicles. The authors noted that by employing an adaptive sample selection strategy, generated images that deviate from the real distribution can be filtered out. As a result, the augmented dataset achieved a 3.84% improvement in mAP as detected by Faster R-CNN, demonstrating a satisfactory enhancement effect [28].

To enhance the accuracy of traffic sign identification, CHRISTINE DEWI et al. synthesized fresh samples by leveraging the distinct benefits of DCGAN, LSGAN, and WGAN for data augmentation. The author points out that in DCGAN, substituting the full connection layer with the convolution layer can better capture the local characteristics of the object. To eliminate the problem of gradient disappearance, batch normalizing technology is added. Different neural networks use a variety of activation functions, including Adam optimization, ReLU, and LeakyReLU. The LSGAN model incorporates a least squares loss function to generate better images, as well as ReLU and Leaky ReLU parameters in the generator and discriminator. To reduce gradient vanishing in WGAN, the Wasserstein distance is utilized, which enhances training stability [29]. To address the challenge of using StyleGAN to generate traffic light sequences with alternating on–off changes, Danfeng Wang et al. designed a conditional style-based TLGAN model. This model utilizes style mixing to separate the background and foreground of traffic lights and introduces a new template loss to force the model to generate traffic light images with the same background but different categories, thereby addressing the issue of imbalanced flashing traffic light data and promoting the model’s generalization ability. This improvement is worth further exploration [30]. In addition, when performing data augmentation, a large dataset needs to be computed, which requires powerful computing power. Training high-quality deep learning models necessitates a substantial amount of annotated data, which significantly increases the cost and time required. To address such issues, Balaji Ganesh Rajagopal et al. conducted research on data augmentation for lightweight road perception pipelines. Firstly, the sim2real technology is applied to convert the semantic segmentation labels generated by the Cityscapes dataset into realistic and diversified street view images. Secondly, based on the proposed hybrid CycleGAN architecture (incorporating superpixel classifiers in the generator and lightweight SVM classifiers in the discriminator), the computational complexity, resources, and time cost of generating images can be reduced, resulting in higher-quality synthetic road images [31].

#### 3.2.2. Challenges and Innovation Strategies of GAN

GAN is a powerful deep learning model that performs well in synthesizing images, but it also faces challenges such as unclear generated images, unstable training, and pattern collapse. The specific details are shown in Table 3.

To encourage GAN to generate images that infinitely approximate real samples, many scholars actively explore its research in traffic scene data and propose innovative response strategies to varying degrees. In Intelligent Transportation Systems, one of the important sources of traffic scene data is point cloud data collected by various sensors, but it has limitations such as large-scale data, high annotation complexity, and poor data quality collected under extreme weather conditions. In response to such issues in Autonomous Driving, Honghui Yang et al. proposed a general self-supervised learning paradigm UniPAD that includes a modality-specific encoder and volume rendering decoder. The method adopts a memory efficient ray sampling strategy to reduce training costs and improve training accuracy. For point cloud data, use 3D backbone network to extract features. Multi-view images utilize 2D backbone networks to extract image features, which are then mapped to 3D space for storage and processing in voxel form. On the other hand, by introducing masking strategies for data augmentation, the goal of selectively masking some inputs to learn more effective features can be achieved. Through experiments, it has been proven that UniPAD fusion has significant effects on multi-modal data, performs well in cross-modal tasks, and increases mAP by 3–5% in 3D object detection tasks. This method avoids the pattern collapse problem that GANs are prone to while maintaining consistency in the generated content space [32].

In addition, the balanced training of the generator and discriminator during the operation of GAN is crucial for the quality of image generation. However, most models are prone to overfitting during training, where the discriminator can accurately distinguish between real and generated data, resulting in high generator losses and poor image quality. Based on this, Yao Gou et al. designed a single-sided mapping multi-feature contrastive learning method for unpaired image-to-image translation to enhance the performance of the discriminator, solving the problem of model collapse. In addition, the author explores and applies the feature information of the discriminator output layer to construct a highly applicable contrasting loss MCL, thereby improving the quality of traffic scene synthesized images [33]. In addition to the discriminator, George Eskandar et al. were inspired by SemanticStyleGAN, which synthesizes images in a combined manner, and proposed Urban StyleGAN which performs category grouping in the pre-training stage to limit the number of local generators. To promote the controllability of image details, the author applied principal component analysis (PCA) in the low-dimensional disentanglement S-space of each category, and validated it on the Cityscapes and Mapillary datasets, generating more controllable and realistic images, optimizing the learning efficiency of the generator, and improving the representation ability of the latent space [34].

### 3.3. Data Augmentation Based on Diffusion Model

#### 3.3.1. Related Work

The proposal of GAN has played a revolutionary role in the development of the generative model field, and the rise of the diffusion model has once again promoted the rapid progress of this field. The diffusion model is inspired by non-equilibrium thermodynamics, and its theoretical basis can be traced back to the diffusion process in physics. The model is composed of forward and reverse diffusion processes. Markov chain is used to simulate the transformation of data from Gaussian noise to real distribution. Because of its unique and brand-new data generation method, high fidelity, and controllable sample generation, it has become the focus of academic and industrial circles in image synthesis, video generation, intelligent monitoring, and other fields.

The core technology of the forward diffusion process is a random process, as shown in Figure 4 (transition from the step 0 to step 10).

The forward process starts from the input image *x*_0_ and, through t iterations, gradually generates images *x*_1_, *x*_2_, …, *x*_t_; $q\left(x_{t}|x_{t-1}\right)$ is a transition distribution. The forward process ($q\left(x_{1:T}|x_{0}\right)$) is a Markov chain, which can be expressed as the following equation [2]:(2)$$q\left(x_{1:T}|x_{0}\right)=\prod_{t=1}^{T}\;q\left(x_{t}|x_{t-1}\right)$$(3)$$q\left(x_{t}|x_{t-1}\right)=N\left(x_{t};\sqrt{1-\beta_{t}}x_{t-1},\beta_{t}I\right)$$

In this process, random noise is gradually added to the pure data without noise through parameters $\beta_{t}$, resulting in the original information being gradually blurred until it is almost completely submerged in the noise. The goal is to transform the original data distribution into standard Gaussian distribution. This process deeply analyzes and accurately captures the internal structure of the data, provides rich and diverse data samples for model training, and enhances the generalization ability of the model. Specifically, the high-noise state after diffusion is the origin of the reverse diffusion process. The round-trip cycle of forward and reverse processes makes the model outstanding in the task of understanding the dynamic behavior of complex systems.

The reverse diffusion process is a denoising process from step 10 to step 0 in Figure 3. It is a Markov chain in the opposite direction, and the goal is to restore the pure Gaussian distribution to the clear data distribution. The process depends on a parameterized neural network, which is trained to identify and eliminate noise, and gradually analyze the real signal hidden under randomness. Reverse diffusion involves learning the probability distribution of an inverse process, which is adjusted based on the output of the previous step, to gradually reduce the noise while retaining or even enhancing the effective structure and features. This process usually depends on variational inference and the theory of fractional differential equations. The reverse process (joint distribution $p_{\theta }(x_{0:T})$ can be expressed as [2]:(4)$$p_{\theta }\left(x_{0:T}\right)=p\left(x_{T}\right)\prod_{t=1}^{T}\;p_{\theta }\left(x_{t-1}|x_{t}\right)$$(5)$$p_{\theta }\left(x_{t-1}|x_{t}\right)=N(x_{t-1};\mu_{\theta }\left(x_{t},t\right),\sum_{\theta }\left(x_{t},t\right))$$

The reverse process starts with $p\left(x_{T}\right)=N\left(x_{T};0,I\right)$ and gradually generates data through the learned Gaussian transfer. The model learns the parameters of the noise distribution through the parameter *θ*, where $\mu_{\theta }\left(x_{t},t\right)$ and $\sum_{\theta }\;(x_{t},t)$ represent the predicted mean and variance, respectively. This allows for gradual denoising and image restoration.

#### 3.3.2. Challenges and Innovation Strategies of Diffusion Model

On the other hand, due to issues such as long training times, slow sampling speeds, and the inability to obtain competitive log-likelihood in the diffusion model, Alex Nichol et al. proposed an improved diffusion probability model. This model, known as the Improved Diffusion Probability Model (IDDPM), aims to address these problems by increasing the number of residual blocks and channels in order to extract a more comprehensive feature distribution and reduce computational complexity. The author discovered that by learning the variance of the reverse diffusion process, the number of required sampling steps can be reduced, resulting in a more efficient sampling process without compromising the quality of the generated data. Additionally, the model incorporates cosine noise scheduling, which allows for smooth adjustments and the addition of noise to improve the representation of high-frequency details [35]. It is worth noting that by imitating the potential distribution of real data, reverse diffusion can enhance the creativity of the model and creatively synthesize new data in the absence of direct samples. This is especially suitable for data augmentation such as small samples, unbalanced categories, rare traffic scenes and image inpainting.

In terms of condition generation, Shih-Yu Sun et al. addressed the issue of a lack of traffic violation video datasets. Firstly, they preprocessed the image to remove vehicles and obtain a clear road image. Secondly, they used a diffusion model to insert vehicles into the road and create abnormal behavior scenes. Finally, these abnormal scenes were combined to generate violation video data, which was then used to train the traffic violation detection system. Through the use of YOLO, an accuracy of approximately 97.36% was achieved, effectively validating the accuracy of the vehicle information in the generated videos and enhancing the system’s robustness [2]. To construct more fine-grained and diverse traffic scenes, Jack Lu and his team employed the proposed SceneControl to generate data samples tailored to specific scenarios. This was achieved by leveraging a highly expressive diffusion model trained on real-world data—capturing elements such as the behavior and speed of designated individuals or vehicles—while integrating a flexible guided sampling mechanism. In addition, SceneControl also generates high-fidelity traffic scenes without conditions to achieve the performance of minimum shortest distance JSD and collision rate. It is efficient and realistic, showing that it has significant advantages over the baseline method in simulating the interaction of participants [36].

In terms of image inpainting and editing of traffic scenes, the diffusion model has strong modeling ability and can achieve excellent image inpainting capability, making it a widely used technique in natural image processing. Building on this, Jiangtong Tan et al. utilized the diffusion model as an auxiliary training mechanism. By incorporating intermediate noise extraction and bottleneck features (H-space features), they were able to overcome the slow reasoning speed of previous methods and obtain high-quality traffic scene images with enhanced granularity and semantic perception ability. This has significant practical applications [37]. There are numerous roads in the world, and ensuring safety should be the top priority. In order to decrease the frequency of traffic accidents and alleviate the burden on traffic management departments, Sumit Mishra et al. have developed an innovative technology for designing safe roads using image inpainting. This technology analyzes the details of accident-prone scenarios to identify unsafe features and then utilizes a deep learning classifier based on ABM, combined with manual feature division. Based on this analysis, the distribution of unsafe characteristics in accident-prone areas is repaired using a diffusion model. The results demonstrate that the classification probability decreases by an average of 11.85% when using the Squeezenet-ABM model. Additionally, a saliency enhancement strategy has been implemented to improve visual saliency through the use of a saliency mask. This includes changing the chromaticity of traffic signs, markings, etc., while also considering human factors [38]. In addition, due to the requirements of real-time, efficiency and computational portability in the process of automatic driving, the lightweight diffusion model came into being. Melike Şah et al. developed a lightweight convolutional neural network model with a processing image size of 32 × 32 × 3, which is 49 times smaller than most network processing sizes. This model includes three different structures, each trained on a different dataset: the original vehicle dataset, a partial diffusion mask dataset, and a complete diffusion mask dataset. The goal of this model is to alleviate the burden on regulatory authorities, public security departments, and other departments by improving the efficiency of vehicle re-recognition. By incorporating diffusion image mask technology, the model is able to learn vehicle feature distribution from multiple angles and directions. The integration of the three network trainings results in high recognition accuracy, surpassing many large-scale pretraining networks [39].

To thoroughly examine the benefits, drawbacks, and suitability of various enhancement techniques from various perspectives, a comparison Table 4 has been included. This table includes parameters such as resolution, computational cost, and training stability.

### 3.4. Composite Data Augmentation

While image transformation, GAN generation models, and diffusion models have greatly improved the quality of traffic visual data, they also have their limitations. As a result, synthetic data augmentation methods are essential. These methods can address the shortcomings of other data augmentation techniques, offering targeted solutions, broad coverage, and effective implementation. In order to address the infrequency of pedestrian posture in traffic accidents, Bo Lu et al. utilized the CARLA simulation platform to gather a larger sample size of accident scene data by incorporating various weather, city, and time variations. Additionally, they employed a combination of physical simulation and synthetic data techniques, incorporating boundary value sampling and genetic algorithms to enhance the diversity of pedestrian postures. This approach also integrated closed-loop optimization strategies, resulting in the development of a comprehensive traffic accident scene data augmentation method known as VCrash [40]. In addition, in order to introduce richer pedestrian posture data samples, enhance the implementation effect of pedestrian detector detection tasks, and improve recognition efficiency, Yunhao Nie et al. proposed a pedestrian data augmentation method with controllable posture in turning situations. This method is based on confidence scores and adjusts the proportion of human postures within different confidence intervals to achieve fine-grained control of posture distribution. At the same time, it greatly improves the detection performance of the detector and is superior in accuracy, recall, and F1 score [41].

To effectively address the challenge of obtaining diverse and hazardous pedestrian data in real-world scenarios, simulation environments such as CARLA and the comprehensive traffic accident scene data augmentation method VCrash can be utilized to generate controllable and diverse pedestrian risk scenarios. This can be achieved by classifying the technical paths and providing actionable practical processes.

The first classification is to perform domain randomization during simulated training. In simulation environments such as CARLA, massive and diverse data is generated by setting different physical attributes. The simulation environment has high efficiency and controllable processes. To improve the diversity of pedestrians in specific scenarios, the second category is divided into two parts: synchronously generating multimodal data in the simulation environment and maintaining semantic consistency through cross-modal feature alignment. Finally, for those extremely dangerous scenarios that are difficult to obtain, a closed-loop reinforcement learning enhanced classification method is adopted to construct an agent (pedestrian) and environment (vehicle) real-time feedback loop, enabling it to approach highly challenging edge scenes. After defining the category, the following steps can be executed to generate diverse pedestrian poses: building a simulated environment and a parametric pedestrian model, designing randomized parameters to account for environmental disturbances and varying motion states, configuring data rendering and acquisition pipelines, and finally training pose estimation models to achieve comprehensive posture coverage (first-level classification).

Based on domain randomization, simulation data generation can be achieved by following these steps: First, establish a connection with the CARLA simulator. Then, randomly configure the simulation environment using “carla.WeatherParameters”. Next, determine the number of pedestrians using a random integer function: “num_pedestrians = random.randint(a, b)”, and randomly select spawn points from all available locations on the map to achieve a randomized distribution of pedestrian density. During the rendering phase, simulate real sensor noise by injecting randomized noise through the camera blueprint attribute setting: “camera_blueprint.set_attribute(‘noise_intensity’, str(random.uniform(NOISE_MIN, NOISE_MAX)))”. Using the sensor data callback function “def sensor_callback(image, data_dict): “, synchronously collect and store both image and annotation data to ensure semantic consistency. Finally, to evaluate the effectiveness of the generated data, introduce domain adaptation techniques to train and optimize the model. Performance is assessed through a dual evaluation framework involving both simulated and real-world scenarios. Using the mAP@50 metric, compare the performance improvement of models trained on real data versus those trained on augmented simulation data when tested on unseen real-world datasets.

Regarding pedestrians, Huiyong Wang et al. designed road monitoring data augmentation methods Mask-Mosaic and Mask-Mosaic++ to address the issue of improving the generalization ability of instance segmentation models based on pedestrian attire in small datasets and speeding up training in large datasets during road monitoring. Additionally, they created a multitasking system that can recognize both clothing types and colors. The results of their experiments showed that Mask-Mosaic++ improved recognition accuracy by 12.37% and 6.17% compared to the original data training model in smaller instances and under varying degrees of occlusion. This approach provides valuable insights for implementing object tracking in the future [42]. Paul M. Torrens et al. used the reverse augmentation method to simulate real people as pedestrians to enhance the dataset. To make the simulation results closer to reality, the author incorporated field research and utilized high-detail models to parameterize virtual elements such as streets and plants. They also combined virtual geographic environments (VGE), virtual reality environments (VRE), and geographic simulation, taking into consideration how the simulated space and phenomena in virtual scenes would map to physical behaviors in the real world, introducing the concept of geographic crossing. By extracting pedestrian features from real data and applying them to the simulation, the author was able to obtain real natural interaction information, resulting in low-cost and high-performance generation strategies. This has the potential to greatly benefit the development of intelligent monitoring, autonomous driving, and intelligent transportation [43].

However, pedestrians, vehicles, traffic signs, and other objects are all small-sized objects in the overall traffic scene. When using an augmented dataset of small object data for model training, the resulting detection accuracy is often lower compared to that of large objects. To address this issue, Brais Bosquet et al. proposed a new network architecture called DS-GAN (a downsampling generative adversarial network), which is designed to generate small objects from large-sized objects in order to improve the accuracy of small object detection. By using the optical flow method to identify suitable positions and combining it with image restoration and blending techniques, the generated small objects are inserted into the scene to increase diversity. The author indicates that this synthetic data augmentation method improves the average precision (AP) by 11.9% and 4.7%, respectively, on the UAVDT dataset when the IoU threshold is set to 0.5, effectively addressing the issue of low accuracy in small object detection in traffic scenes [44]. In addition, for the problem of class imbalance, A.S. Konushin et al. studied three methods including “Pasted”, “CycleGAN”, and “StyledGAN” to synthesize rare traffic sign data, improving background consistency, coordination, and diversity. Additionally, an improved variational autoencoder (VAE) was utilized to select the optimal position for the newly generated images. The experiments showed that the highest accuracy of 94.11% was achieved when using an optimized classifier to classify rare and common categories. Furthermore, when only training on the generated data, the classification accuracy improved by 12.48%, demonstrating excellent performance [45]. Jingyi Shi et al. proposed the use of parameters to regulate the object pasting rate and scaling rate in order to shift the focus of multi-image fusion towards object cropping and pasting. This approach enriches the diversity and scale variation of traffic sign small object data. The results of testing on the CTSD dataset showed an increase in mAP0.5 and mAP0.5: 0.95 by 3.5% and 2.5%, respectively, compared to those achieved with mosaic data [21].

### 3.5. Selection Criteria for Four Data Augmentation Techniques

Image transformation-based data augmentation is widely adopted by researchers who require fast sample processing, diverse transformations, and flexible application. This approach proves especially valuable in high-performance tasks such as object detection and segmentation, as it lowers the threshold for practical use. Within this context, GAN demonstrates strong capabilities in generating images and enriching datasets, particularly under complex traffic scenarios. These advantages help address common challenges such as limited dataset size and insufficient image quality. Beyond traffic-related research, GAN has also been applied in areas such as image style transfer, image restoration, and video generation. Similarly, diffusion models excel at producing highly realistic images through simple loss functions during training. They can also translate text into images, which has expanded their adoption across domains including art, graphic design, film animation, media, and gaming. When a single augmentation method cannot meet the demands of complex tasks, composite augmentation strategies have emerged as effective alternatives. These combined approaches significantly enhance model generalization and robustness in challenging scenarios [46].

GAN and diffusion models are now extensively used in data augmentation research due to their exceptional performance in several key areas:(i)Enhancing visual authenticity in transportation scenarios: Existing traffic datasets often suffer from imbalanced categories and limited diversity. GANs address this by capturing fine-grained details from available data, using adversarial training between generators and discriminators to synthesize realistic images. They can also generate temporally consistent data with dynamic characteristics, improving continuity and overall model performance. Diffusion models, on the other hand, add noise to data and train neural networks to iteratively restore images. This process allows them to closely match real data distributions, generating diverse and structurally accurate traffic scene images. Their modular design further supports multi-modal inputs and offers strong structural flexibility, enabling applications in fault detection and sample synthesis.(ii)Prevalence in recent literature: A Google search with the keywords “GAN,” “Diffusion Model,” and “Traffic Scene” yielded approximately 16,700 and 16,900 cited results since 2021, respectively. This reflects their strong popularity and widespread adoption in traffic visual scene research, underscoring their practical value.(iii)Diversity in generation methods: GANs rely on adversarial synthesis with minimal architectural constraints, enabling a wide variety of model variants. Diffusion models operate through the iterative process of “adding noise and then removing it,” which can be guided by multiple conditional mechanisms. This design offers high flexibility and controllability, making them particularly powerful for structured generation tasks.

### 3.6. Comparison of the Challenges Between GAN and Diffusion Models

Although GAN and diffusion models have achieved remarkable results in generating traffic visual element images, both approaches continue to face significant challenges, as summarized in Table 5:

### 3.7. Multimodal Enhancement in GAN and Diffusion Models

When the enhanced model processes complex data with multimodal traffic visual information, the ordinary real linear layer cannot effectively improve the model’s ability. In addition, the number of parameters is too large. At this point, it is necessary to introduce hypercomplex linear layers (HCL) such as quaternions. These layers replace ordinary linear layers or 1 × 1 convolutions. They map data to high-dimensional space, enhancing the expressive and modeling performance of GAN and diffusion models in capturing multimodal distribution features. Wen Shen et al. addressed the issues of performance degradation and poor robustness against rotation attacks in traditional neural networks for processing 3D point cloud data in autonomous driving. They represented each 3D point and intermediate layer feature using quaternions to compensate for the rotation-equivariance and permutation-invariance problems [53].

HCL based on quaternion algebra can generally be expressed as *a* + *bk* + *cj* + *di* (*a*, *b*, *c*, *d* are real numbers, *i*, *j*, *k* are imaginary units and *i*^2^ = *j*^2^ = *k*^2^ = *ijk* = −1). In neural networks, the core operations consist of quaternion multiplication and quaternion convolution. Firstly, convert the input data into quaternions, then perform quaternion multiplication, and after passing through the activation function, perform output conversion. Here is a simple quaternion linear layer code snippet based on PyTorch 2.4.1+cu118:

import torch

import torch.nn as nn

class QuaternionLinear(nn.Module):

       def __init__(self, in_features, out_features):

              super(QuaternionLinear, self).__init__()

              self.in_features = in_features

              self.out_features = out_features

              self.w_r = nn.Parameter(torch.Tensor(out_features, in_features))

              self.w_i = nn.Parameter(torch.Tensor(out_features, in_features))

              self.w_j = nn.Parameter(torch.Tensor(out_features, in_features))

              self.w_k = nn.Parameter(torch.Tensor(out_features, in_features))

              self.reset_parameters()

       def reset_parameters(self):

nn.init.xavier_uniform_(self.w_r)

             nn.init.xavier_uniform_(self.w_i)

             nn.init.xavier_uniform_(self.w_j)

             nn.init.xavier_uniform_(self.w_k)

def forward(self, x):

      xr, xi, xj, xk = torch.chunk(x, 4, dim = −1)

             r_output = xr@ self.w_r.T- xi@ self.w_i.T-xj@ self.w_j.T-xk@ self.w_k.T

             i_output = xr@self.w_i.T+xi@ self.w_r.T+ xj@ self.w_k.T- xk@ self.w_j.T

             j_output = xr@self.w_j.T-xi@ self.w_k.T+xj@ self.w_r.T+xk@ self.w_i.T

             k_output = xr@self.w_k.T+xi@ self.w_j.T-xj@ self.w_i.T+xk@ self.w_r.T

             output = torch.cat([r_output, i_output, j_output, k_output], dim = −1)

             return output

To provide a more detailed explanation of the differences in computational complexity and activation functions between the real valued layers and the hypercomplex layers, the following Table 6 is developed for comparison:

When using hypercomplex layers for deep learning tasks, conventional loss functions such as cross-entropy remain unchanged in the hypercomplex setting, since these functions operate on scalar outputs that are independent of the underlying parameter representation. The distinction arises in the backpropagation stage, where gradient computations are carried out using multidimensional algebra that respects the algebraic rules of the hypercomplex domain. Optimizers such as SGD or Adam are then applied in a component wise manner across the real, imaginary, or higher order components of the hypercomplex parameters. For instance, in the case of quaternion-valued networks, each weight parameter is represented by four components. During training, the loss gradient with respect to each component is computed, and Adam updates are performed separately on the real, *i*, *j*, and *k* components. This procedure preserves the familiar optimization dynamics of Adam while ensuring that the structural relationships within the hypercomplex representation are maintained. By doing so, the framework extends conventional deep learning techniques into richer parameter spaces without altering the fundamental principles of loss evaluation and optimization.

### 3.8. Summary

In this section, we have provided a comprehensive introduction to various data augmentation methods. In order to examine the specific impact of each method on model performance, we have presented multiple deep learning models with varying complexities and parameter counts. The effectiveness of these data augmentation methods is evaluated in Table 7.

The results of the experiment demonstrate that different data augmentation techniques have varying levels of success on different datasets and models. In cases where computing resources are limited, a simple image transformation method can serve as a basic enhancement strategy. However, for those seeking to generate highly realistic images and increase data diversity, GAN can be utilized to address the issue of data scarcity. After conducting a comparison, it was found that LSGAN outperforms other methods, with more stable training and higher quality image generation. The performance improvement of the diffusion model is higher, indicating that it has better realism and diversity compared to GAN. However, it requires a substantial amount of computing resources and technical support. Additionally, a combination of data augmentation methods can be tailored to the specific application field, enhancing the model’s generalization ability in real-world scenarios and increasing its effectiveness.

This survey distinguishes itself by bridging classical augmentation techniques with emerging generative models such as diffusion and composite approaches, systematically comparing their computational trade-offs, dataset-specific performances, and implementation feasibility in Intelligent Transportation Systems from the perspective of traffic visual elements such as people, vehicles, traffic lights, and roadblocks. The resulting synthesis serves as both a reference and a roadmap for future data augmentation research in autonomous driving.

## 4. Datasets and Evaluation Index

There are many visual elements involved in transportation, and there are also numerous experts and scholars studying transportation scenes. However, few have organized and analyzed the relevant datasets. In the following sections, a summary of the various traffic visual elements will be conducted.

### 4.1. Datasets Related to Traffic Visual Elements

#### 4.1.1. Traffic Sign Datasets

In the auto drive system, traffic signs, as key structural visual elements, provide indispensable environmental semantic information for vehicle decision-making. The relevant representative dataset information is shown in Table 8.

#### 4.1.2. Traffic Light Datasets

In addition to traffic signs, traffic lights are also one of the fundamental and multi-dimensional visual elements in autonomous driving scenarios. The relevant representative dataset information is shown in Table 9.

#### 4.1.3. Traffic Pedestrian Datasets

Pedestrians are the most important participants in the traffic environment, and in traffic accidents, they belong to the vulnerable party. Therefore, accurate and timely detection and identification of pedestrians is the primary prerequisite for achieving vehicle safety. The relevant representative dataset information is shown in Table 10.

#### 4.1.4. Vehicle Datasets

Vehicles are the most involved component of the transportation system. The accurate and real-time detection of vehicle elements is directly related to the safety and interactive capabilities of autonomous driving. The relevant representative dataset information is shown in Table 11.

#### 4.1.5. Road Datasets

The road is an important carrier for vehicle operation and a visual representation of traffic rules. The correct detection of road signs provides rule constraints for autonomous driving. The relevant representative dataset information is shown in Table 12.

#### 4.1.6. Traffic Scene Datasets

The traffic scene integrates all the information of the traffic visual elements and provides the auto drive system with comprehensive understanding information below, which is of the highest importance. The relevant representative dataset information is shown in Table 13.

#### 4.1.7. Process for Selecting and Filtering Datasets

To provide an operational dataset evaluation framework and guide relevant researchers to complete the entire process from requirement analysis to final decision-making, a process for selecting and filtering datasets has been developed as shown in Figure 5.

#### 4.1.8. The Gaps in Scarce Transportation Scenarios

Although nearly 40 datasets provide valuable resources for standard analysis and exploration, such as CrowdHuman for occlusion, NightOwls for nighttime, and D^2^-City for challenging weather, they remain limited in terms of lighting diversity and extreme weather coverage.

(i)Low-illumination environments: Most datasets, including TUD Brussels Pedestrian, Cityscapes, JAAD, Elektra, Udacity, NYC3DCars, GTSRB, and Detection Benchmark, focus on daytime conditions and contain few nighttime or varied weather samples. Oxford Road Boundaries incorporates seasonal and lighting variations but still lacks data for heavy rain, dense fog, or pure nighttime. Similar gaps exist in the TME Motorway Dataset and RDD 2020. In contrast, CTSDB, TT 100K, and STSD capture broader weather and lighting conditions. Mapillary Traffic Sign Dataset adds seasonal, urban, and rural variability, while NightOwls specializes in nighttime pedestrian detection across multiple European cities.(ii)Harsh weather adaptation: Many datasets lack sufficient fog, rain, or snow samples, restricting model robustness in real traffic scenarios. For instance, CCTSDB suffers from this limitation. By comparison, D^2^-City and ONCE contain extensive challenging weather conditions, supporting domain adaptation research. Bosch Small Traffic Lights Dataset also addresses this issue, offering diverse weather and interference for signal detection tasks.(iii)Small, occluded, and complex samples: Datasets such as LISA Traffic Sign, DriveU Traffic Light, CeyMo, and CrowdHuman attempt to address scarcity in small targets, occlusion, and cluttered scenes. LISA provides detailed annotations including occlusion. DriveU emphasizes small pixel objects. CeyMo introduces nighttime, glare, rain, and shadow. CrowdHuman enriches severe occlusion scenarios. Other datasets fill additional gaps, such as Chinese City Parking (parking lot environments), PANDA (tiny distant objects), and CeyMo (unique perspectives). These contributions improve coverage but remain incomplete and unbalanced.(iv)Regional and special scenarios: Certain datasets focus on geographic diversity and context-specific challenges. KUL Belgium Traffic Sign emphasizes sign variation, while RTSD adds extreme lighting, seasonal diversity, and Eastern European symbols absent in mainstream datasets. LISA Traffic Light covers day and night variations, complemented by LaRA’s signal videos. PTL combines pedestrians and traffic lights. ApolloScape and BDD100K expand coverage to multiple cities, environments, and weather types. Street Scene captures complex backgrounds with vehicles, pedestrians, and natural elements. Highway Workzones highlights construction zones with cones, signs, and special vehicles.

Overall, most datasets address gaps only in isolated aspects, which introduces data bias and cognitive defects in autonomous driving systems. These limitations hinder the handling of long tail scenarios, leading to reliability issues such as missed or false detections, failures in nighttime driving, and regional restrictions. To overcome these weaknesses, targeted data augmentation strategies are needed to enhance diversity, sufficiency, and category balance, thereby improving model robustness and generalization.

### 4.2. Evaluation Index

To better evaluate the quality of generated images, visual observation alone is insufficient to discern differences in detail distribution and changes in diversity. Therefore, targeted indicators need to be used for measurement. Based on the details, textures, colors, shapes, background complexity, and other characteristic information of traffic visual elements, the following evaluation indicators are selected for quantification.

#### 4.2.1. Inception Score (IS)

The Inception Score quantifies the performance of a generative model by evaluating the diversity and clarity of the generated images, with a higher value indicating better performance. This score is based on the Inception Net-V3 model and uses a pre-trained Inception network to classify the generated images and obtain a predicted probability distribution. However, it heavily relies on classifiers and is an indirect method for evaluating image quality, which may overlook the distribution of real data. Additionally, if there is a class imbalance in the generated images, the Inception Score may not accurately represent the overall quality of the generated images. The specific calculation process is shown in the following equation:(6)$$IS(G)=exp\left(E_{x\sim p_{g}}D_{KL}(p(y|x)\|p(y))\right)$$
where $x\sim p_{g}$: Images generated from the generator; p(y∣x): Predict probability distribution; $D_{KL}$: KL divergence, which is used to measure the degree of approximation of p(y∣x) and p(y); p(y): Edge probability, for all generated images, calculate p(y∣x) and analyze the average of all vectors [99].

#### 4.2.2. Fréchet Inception Distance (FID)

FID directly quantifies the difference between the generated image and the real image by measuring the distance between two multivariate normal distributions. A smaller value indicates a better match. Its main concept is to utilize a pre-trained Inception Net-V3 network to extract features. However, this approach heavily relies on pre-trained networks, and the extracted features may not be optimal for specific tasks. This can lead to overfitting and high computational complexity. The specific calculation process is shown in the following equation:(7)$$FID={\left\|\mu_{r}-\mu_{g}\right\|}^{2}+T_{r}(\sum_{r}+\sum_{g}-2{(\sum_{r}\sum_{g})}^{1/2})$$
where $\mu_{r}$, $\Sigma_{r}$: The feature mean and covariance matrix of real images; $\mu_{g}$, $\Sigma_{g}$: The feature mean and covariance matrix of the generated image; $T_{r}$: Trace [100].

#### 4.2.3. Kernel Inception Distance (KID)

KID evaluates the convergence of GAN by calculating the Maximum Mean Discrepancy (MMD) between the generated image and the real image in the Inception network feature space. A smaller value indicates better convergence. Its core calculation formula is shown below.(8)$$KID={MMD}^{2}\left(F_{r},F_{g}\right)$$
where $F_{r}$ and $F_{g}$: Inception features from real images and generated images.

${MMD}^{2}$ refers to the squared maximum mean difference. Consider samples $X={\left\{x_{i}\right\}}_{i=1}^{m}$ from distribution $F_{r}$ and samples $Y={\left\{y_{i}\right\}}_{i=1}^{n}$ from $F_{g}$. MMD’s unbiased estimation is shown in the following equation.(9)$${MMD}^{2}=\frac{1}{m(m-1)}\sum_{i\neq j}^{m}k(x_{i},x_{j})+\frac{1}{n(n-1)}\sum_{i\neq j}^{n}k(y_{i},y_{j})-\frac{2}{mn}\sum_{i=1}^{m}\sum_{j=1}^{n}k(x_{i},y_{j})$$
where k(·,·) is the kernel function. The commonly used polynomial kernels is $k(x,y)={\left(\frac{1}{d}x^{T}y+1\right)}^{3}$. *x* and *y* are feature vectors extracted from the Inception network. *d*: Dimensions of representation.

KID is similar to FID in that it can directly evaluate the differences between generated images and real images. It uses a third-order polynomial kernel and also compares skewness in the process of comparing mean and variance. In some cases, it is more robust to changes in sample size [101].

#### 4.2.4. Jensen–Shannon Divergence

JS divergence is a measure of similarity between probability distributions, used to assess the similarity between generated images and real images. It is a symmetric version of KL divergence, with values ranging from 0 to 1. The specific calculation process is shown in the following equation:(10)$$JS\left(P\left\|Q\right.\right)=\frac{1}{2}KL\left(P\left\|M\right.\right)+\frac{1}{2}KL(Q\left\|M\right.)$$
where *P*, *Q* are the given two probability distributions; *M* is the average distribution of *P* and *Q* [102].

#### 4.2.5. Peak Signal-to-Noise Ratio (PSNR)

PSNR is based on the concept of signal and noise, reflecting the ratio between blurry areas, distortions, and other artifacts (noise) that appear in the real image (signal) and the generated image. The larger its value, the smaller the loss and the better the quality of the generated image. The calculation process is simple and suitable for large-scale image processing tasks, but it cannot capture the details and textures of the image, so there may be situations where the results are misjudged. The specific calculation process is shown in the following equation:(11)$$PSNR=10\cdot \log_{10}\left(\frac{{MAX}^{2}}{MSE}\right)$$
where MSE: Mean squared error; MAX: The maximum possible pixel value in the image.

#### 4.2.6. Structural SIMilarity (SSIM)

SSIM is more user-friendly than PSNR; for images *x* and *y*, brightness ($l\left(x,y\right)=\frac{2\mu_{x}\mu_{y}+c_{1}}{\mu_{x}^{2}+\mu_{y}^{2}+c_{1}}$, $c_{1}={\left(k_{1}L\right)}^{2}$, $k_{1}=0.01$, *L*: range of pixel values), contrast ($c\left(x,y\right)=\frac{2\sigma_{x}\sigma_{y}+c_{2}}{\sigma_{x}^{2}+\sigma_{y}^{2}+c_{2}}$, $c_{2}={\left(k_{2}L\right)}^{2}$, $k_{2}=0.03$) and structural similarity ($s\left(x,y\right)=\frac{\sigma_{xy}+c_{3}}{\sigma_{x}+\sigma_{y}+c_{3}}$, $c_{3}=\frac{c_{2}}{2}$) are used to measure the similarity between them, and the larger the value, the better. $\mu_{x}$ and $\mu_{y}$ are the mean values of *x* and *y*; The variances of *x* and *y* are represented by $\sigma_{x}^{2}$ and $\sigma_{y}^{2}$, respectively; $\sigma_{xy}$ is the covariance of *x* and *y* [103]. The specific calculation process is shown in the following equation:(12)$$SSIM\left(x,y\right)=[l{\left(x,y\right)}^{\alpha }\cdot c{\left(x,y\right)}^{\beta }\cdot s{\left(x,y\right)}^{\gamma }]$$

Usually, α=β=γ=1, then:(13)$$SSIM=\frac{\left(2\mu_{x}\mu_{y}+c_{1})(2\sigma_{xy}+c_{2}\right)}{\left(\mu_{x}^{2}+\mu_{y}^{2}+c_{1})(\sigma_{x}^{2}+\sigma_{y}^{2}+c_{2}\right)}$$

#### 4.2.7. Feature Similarity Index (FSIM)

To simulate the information extraction of the human visual perception system, Lin Zhang et al. proposed the image quality evaluation index FSIM in 2011, which considers both structural and detailed information. The larger the value, the better. The core idea is to measure the quality of generation by capturing phase consistency and gradient amplitude similarity. The specific calculation process is shown in the following equation:(14)$$FSIM(I_{1},I_{2})=\frac{\sum_{x}\emptyset (x)\cdot {SIM}_{G}(x)}{\sum_{x}\emptyset (x)}$$
where ∅(x): Phase consistency function; ${SIM}_{G}(x)$: Gradient amplitude similarity function [104].

#### 4.2.8. Learned Perceptual Image Patch Similarity (LPIPS)

Richard Zhang et al. introduced LPIPS in 2018, which employs deep neural networks to extract advanced features, such as image semantics. This allows for the measurement of image quality by comparing the similarity of these features, with a lower value indicating better quality. This approach is more aligned with the human visual perception system, but its reliance on extracting multiple layers of features results in high computational complexity. The calculation process is represented by the L2 norm ${\left\|\cdot \right\|}_{2}$, as shown in the following equation:(15)$$LPIPS(I_{1},I_{2})=\sum_{l}w_{l}{\left\|\emptyset_{l}\left(I_{1}\right)-\emptyset_{l}\left(I_{2}\right)\right\|}_{2}$$
where $w_{l}$: Weight learned; $\emptyset_{l}\left(I\right)$: Feature extraction function of layer *l* [105].

#### 4.2.9. Gradient Magnitude Similarity Deviation (GMSD)

GMSD evaluates the difference between the generated image and the real image by calculating the different gradient amplitudes produced by different structures in the image during the change process, and the smaller the value, the better. According to the proposed standard deviation pooling concept, the Prewitt filter is used for analysis and calculation, as shown in the following equation:(16)$$h_{x}=\left[\begin{array}{c} 1/3\;0-1/3\\ 1/3\;0-1/3\\ 1/3\;0-1/3 \end{array}\right],\;h_{y}=\left[\begin{array}{c} 1/3\;1/3\;1/3\\ 0\;0\;0\\ -1/3\;-1/3-1/3 \end{array}\right]$$(17)$$m_{r}(i)=\sqrt{{\left(r⨂h_{x}\right)}^{2}\left(i\right)+{\left(r⨂h_{y}\right)}^{2}\left(i\right)}$$(18)$$m_{d}(i)=\sqrt{{\left(d⨂h_{x}\right)}^{2}\left(i\right)+{\left(d⨂h_{y}\right)}^{2}\left(i\right)}$$(19)$$GMS(i)=\frac{2m_{r}\left(i\right)m_{d}\left(i\right)+c}{m_{r}^{2}\left(i\right)+m_{d}^{2}\left(i\right)+c}$$
where $h_{x}$, $h_{y}$: Definition of Prewitt Filter in Horizontal and Vertical Directions; $m_{r}$, $m_{d}$ gradient magnitude; *c*: constant [106].

#### 4.2.10. Deep Image Structure and Texture Similarity (DISTS)

DISTS utilizes convolutional neural networks to comprehensively consider structural and texture similarity, jointly optimize and evaluate the quality of generated images, and simulate HVS systems more comprehensively and accurately. The specific analysis process can refer to the literature [107].

#### 4.2.11. Evaluation Metrics Selection and Analysis of Perceptual and Functional Correlation

The selection of the ten evaluation metrics was carefully designed to address both perceptual quality and functional relevance in traffic applications. On the perceptual side, metrics such as the Fréchet Inception Distance (FID) and Inception Score (IS) were included because they are widely recognized for evaluating generative model outputs in terms of realism, diversity, and overall visual appeal. These are complemented by structural fidelity measures such as the Structural Similarity Index (SSIM), which focus on how well generated or reconstructed images preserve essential visual details compared to reference images. Together, these metrics ensure that the aesthetic and perceptual quality of outputs align with human visual judgment.

At the same time, traffic-related systems demand more than visual plausibility. To reflect this, the framework incorporates task-oriented functional metrics such as mean Average Precision (mAP) and Intersection over Union (IoU). These measures directly assess how well images support critical downstream tasks like object detection, lane segmentation, and recognition, tasks that are central to real world intelligent transportation systems. By grounding part of the evaluation in application level outcomes, the framework ensures that models are not only visually convincing but also effective in supporting decision making processes.

Finally, diversity and robustness indices were included to ensure that model outputs capture the variability present in complex traffic environments. This prevents overfitting to narrow scenarios and supports generalization across different conditions such as lighting, weather, and traffic density. Prior survey studies have consistently emphasized the need to balance perceptual and functional perspectives, and our metric selection reflects this by spanning both families [108]. As such, the chosen set provides a sufficiently holistic evaluation framework, ensuring that models can deliver outputs that are both realistic in appearance and reliable in function.

In order to systematically evaluate the improvement of downstream perception task performance by different data augmentation strategies, a mapping relationship of “Enhancement Methods-Improvement Type-Evaluation Indicator” was constructed as shown in Table 14, providing reference for related research.

### 4.3. Summary

This section provides a summary of approximately 40 commonly used datasets and 10 evaluation metrics in the field of traffic vision. These resources serve as a strong basis for the development and evaluation of models for tasks such as object detection and classification. It is important to note that when conducting model testing, it is crucial to comprehensively evaluate the model’s performance on datasets from various regions and under different collection conditions. Additionally, the diversity of evaluation indicators should be taken into consideration in order to enhance the model’s generalization ability.

## 5. Discussion

### 5.1. Feature Extraction and Semantic Understanding Enhancement

There are various visual elements in transportation, and the background is constantly changing. During the enhancement process, it is easy to focus on object feature extraction and ignore background diversity, resulting in a decrease in the model’s generalization ability. Some researchers use methods such as setting environmental parameters, simulating noise, and replacing to synthesize backgrounds. However, these methods lack contextual semantic information of the scene and cannot accurately represent the physical rules and logical relationships between different regions in the scene. Therefore, it is necessary to strengthen the background’s semantic understanding ability and extract more comprehensive and rich detailed features through multi-data fusion while maintaining consistency between the upper and lower text.

### 5.2. Physical Barrier Information Dataset

Physical barrier information, such as traffic cones and traffic safety barrels is common in real traffic scenarios, but the relevant data sets are relatively scarce, which will lead to autonomous vehicles failing to learn effective information and unnecessary traffic accidents in the process of moving. Therefore, the collection of real-world scenarios for such datasets is also essential. In addition, various simulation platforms should be fully utilized to construct long tail scene datasets such as traffic cones that are “uncommon”. Integrating physical information with deep learning enhances interpretability and physical consistency, which not only improves model performance but also reduces dependence on observed data. Forming a virtuous cycle.

### 5.3. Data Augmentation Strategy for Integrating Diversity and Authenticity of Traffic Visual Scenes

To comprehensively consider the diversity and authenticity of traffic visual elements scenes, consideration should be given to adopt conditional generative models (e.g., class-/weather condition-driven GANs), multi-modal fusion (combining LIDAR or radar), simulation to real pipelines, and reinforcement learning-based augmentation that iteratively refines synthetic content based on downstream model feedback. In addition, for cross domain adaptation and optimization of traffic visual elements in different regions, countries, and weather conditions, should be combined domain randomization, adversarial domain adaptation, self-supervised pretraining across synthetic-real combined datasets, and curriculum augmentation that gradually increases complexity [109]. We believe that to address the challenges of missing extreme scenario samples, insufficient semantic information, and the lack of authenticity and diversity in domain adaptation, cross-domain applications of cross-modal image generation models can be employed. This approach enables the fusion and transformation of different modalities, generating high-value extreme scenario data that directly tackles the core pain point of long-tail data scarcity in autonomous driving.

### 5.4. Generating Efficiency and Computational Cost

Most data augmentation methods require a long time and a large amount of computing resources in the process of generating images, and training requires multiple adjustments. Therefore, the efficiency and low cost of data augmentation have always been urgent problems to be solved. We should focus on lightweight network architecture as a breakthrough point and put effort into improving image generation efficiency. However, most current lightweight network designs primarily trade off efficiency at the expense of visual fidelity, which poses serious risks to the safety of autonomous driving. We recommend that, while considering computational cost as a key metric for evaluation, it is also essential to comprehensively balance model performance by incorporating downstream performance gains such as accuracy and generalization capability.

## 6. Conclusions

This paper presents the first structured synthesis of data augmentation techniques specifically designed for traffic visual elements, a critical yet relatively underexplored domain in autonomous driving perception. By developing a comprehensive taxonomy that integrates transformation-based, GAN-driven, diffusion, and composite methods, we construct a cross-comparative benchmark covering nearly forty datasets and ten evaluation metrics. The novelty of this work lies in its multi-layered analytical framework, which systematically links augmentation strategies to real-world outcomes such as accuracy, mean average precision (mAP), and robustness while examining their computational and semantic trade-offs, illustrating how emerging generative paradigms, particularly diffusion and multimodal composite models, enrich dataset diversity and improve the representation of rare driving scenarios.

Beyond synthesis, this work offers tangible implementation insights. It provides practical guidance for selecting augmentation methods under computational constraints. It also identifies key unresolved challenges, including developing lightweight generative pipelines, enhancing semantic understanding, and establishing physical barrier information datasets. Finally, the study delineates future research directions to strengthen data reliability and cross-domain transferability in intelligent transportation systems.

This study serves as a foundational reference that consolidates current progress while charting a forward-looking roadmap for researchers and practitioners committed to developing resilient, real-world perception models for autonomous driving.

## Figures and Tables

**Figure 1 sensors-25-06672-f001:**
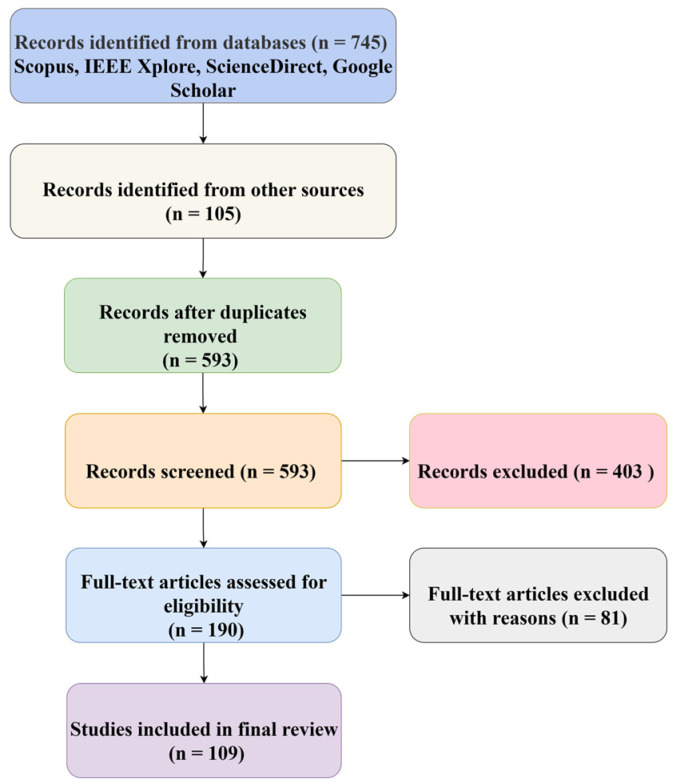
PRISMA flow diagram.

**Figure 2 sensors-25-06672-f002:**
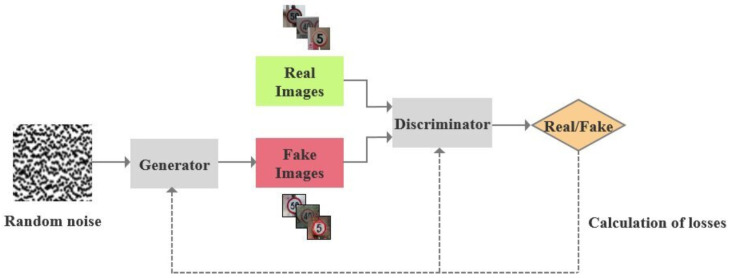
Generative Adversarial Network Structure. It consists of a generator and a discriminator.

**Figure 3 sensors-25-06672-f003:**
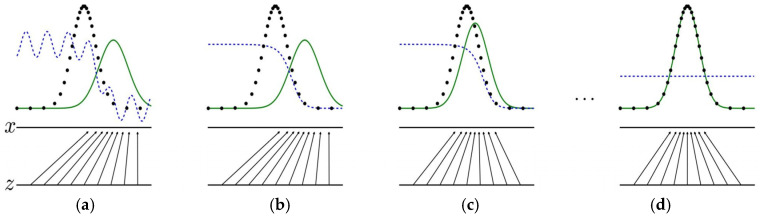
The Iterative Process of GAN. The generator and discriminator engage in a continuous competition, ultimately reaching a Nash equilibrium. (**a**) Initial State; (**b**) Training Discriminator; (**c**) Training generator; (**d**) Final State.

**Figure 4 sensors-25-06672-f004:**
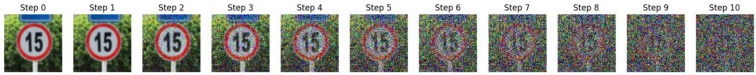
Iterative Process of Diffusion Model.

**Figure 5 sensors-25-06672-f005:**
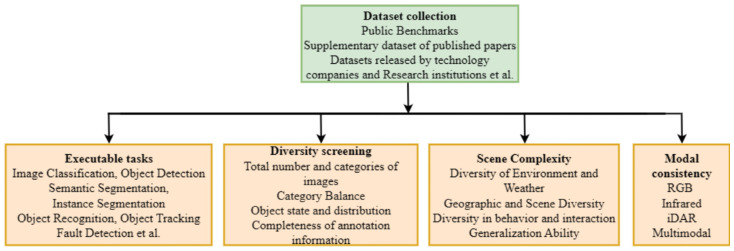
Process diagram for selecting and filtering datasets.

**Table 1 sensors-25-06672-t001:** Simple image transformation method.

Method Category	Specific Method	Describe	Effect	Deficiency
Geometric transformation	Rotation	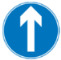 → 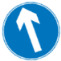	Different angles of simulated objects.	It may lead to the loss of edge information, background filling and other problems.
Scaling	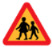 → 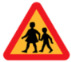	Enhance the recognition ability of the model for different size objects.	The loss of detail information, the introduction of interpolation noise, resulting in image blur.
Translation	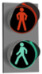 → 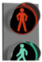	Different locations of simulated objects.	The object position is offset, which affects the correct position information.
Flipping	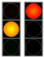 → 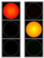	Facilitate the model to learn the characteristics of the object in different directions.	Change the context information of the object.
Random Cropping	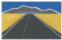 → 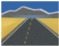	Different perspectives of simulated objects.	Important information may be lost.
Affine Transformation	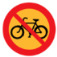 → 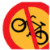	Generate diversified global samples.	High computational complexity.
Color transformation	Brightness	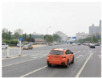 ↓ 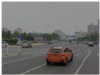	Simulate different light changes.	Brightness imbalance, loss of details.
Contrast	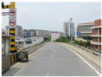 ↓ 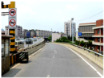	Allows the model to more easily extract key features.	Excessive enhancement may lead to noise amplification and information loss.
Saturation	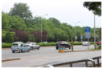 ↓ 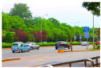	Simulate color performance under different lighting conditions.	Easy to cause color distortion.

**Table 2 sensors-25-06672-t002:** Comprehensive image transformation method.

Methods	Describe	Effect	Deficiency
Shadow or reflection	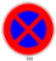 → 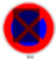	Enhance the realism of light occlusion.	Too large easily leads to difficulty in object feature extraction.
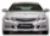 → 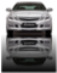	Increase data volume.	Cause semantic errors.
Simulate different weather and light conditions	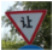 → 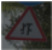	Enhance the adaptability of the model to complex environments	Lack of authenticity.
Noise addition	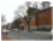 → 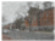	Enhance model robustness.	Cause image distortion.
Fuzzy processing	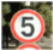 → 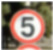	Improve the recognition ability of the model for different definition images.	Affect learning of key features.
Hybrid augmentation	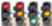 → 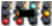	Enrich data.	High computational cost and blurred image labels.

**Table 3 sensors-25-06672-t003:** Summary of problems that are prone to occur during GAN training.

The Resulting Issues	Describe	Solution
Mode Collapse	Due to the generator finding an output that can “deceive” the discriminator, it no longer explores other possibilities, resulting in a high similarity of most of the generated samples.	Introducing regularization techniques: incorporating methods such as L1 and L2 regularization into the loss function to improve the model’s ability to generate diverse samples.
		Adding noise: Injecting random noise into the input or network layer of the generator to promote the diversity of generated samples.
		Gradient penalty: Punishing the excessive gradients of the discriminator to prevent them from becoming too strong during training.
Gradient Vanishing	During the training process, the gradient values decrease layer by layer and gradually disappear, resulting in sluggish or even failed convergence of the neural network. The generator cannot be trained.	Depth the network architecture
		Add residual connection
		Batch regularization
		Change the activation function
		Adjust learning rate
Image Blur	Due to the imbalanced training of the generator and discriminator, the generated images have unclear details, poor quality, and unstable model training.	Optimize network architecture
		Improve loss function
		Training of Balance Generator and Discriminator
		Adjust training strategy
		Adjust hyperparameters
Training Instability	During the training process, when either the generator or discriminator becomes too powerful, it can lead to an imbalance in the entire training process and unstable generation quality.	Optimize network architecture
		Adjust training strategy
		Data preprocessing and augmentation
		Batch standardization
		Adjust hyperparameters
Overfitting	The generator performs well on training data but performs poorly on new, unseen data.	Introducing regularization techniques
		Data preprocessing and augmentation
		Label smoothing
Underfitting	The generator cannot fully learn the distribution of the training data, resulting in its deviation from the real distribution.	Increasing the network complexity
		Adjusting training parameters
		Extending the number of iterations

**Table 4 sensors-25-06672-t004:** Comparison Table of Parameters for Various Data Enhancement Methods.

Comparative Dimension	Image Transformation	GAN	Diffusion Model
Calculation cost	low	moderate	The demand for graphics memory and computing power far exceeds that of GAN.
Resolution	unchanged	high resolution (1024 × 1024)	high resolution (1024 × 1024)
Training stability	very stable	low	relatively stable
Semantic fidelity	low	The generated results are often not fully aligned with the semantic meaning of the text.	High fidelity generation
Small object applicability	low	moderate	The generated samples have rich details and far greater diversity than GAN.
License restrictions	none	Attention should be paid to data licensing, etc.	Attention should be paid to data licensing, etc.
Inference speed	extremely fast	quick	slow

**Table 5 sensors-25-06672-t005:** Comparison Table of Challenges between GAN and Diffusion Models.

Models Name	Challenges	Specific Manifestations
GAN	Training instability, which leads to fluctuations in generation quality, and even result in complete failure of the training process [47].	The fundamental challenge of GAN lies in the difficulty of achieving dynamic balance between the generator and discriminator during adversarial training. While there have been improvements in training strategies and loss function compensation, there is still a lack of conditional control for generating rare scene data based on specific needs, particularly for small object data augmentation tasks. To address this issue, it is necessary to combine GAN with supervised enhancement methods that use labels from training samples to guide data augmentation, or to develop effective solutions for generating super-resolution training data to address the problem of data loss in rare scenes.
	Mode collapse, which refers to the generator starting to generate very limited sample diversity [48,49].	
	Difficulty capturing rare traffic scenarios [50]	
Diffusion Models	High computational complexity. Reference [51] comprehensively reviewed the issues of data loss and multimodality in the field of transportation using diffusion models, pointing out their challenges in terms of efficiency, controllability, and generalization.	The diffusion model is known for its time-consuming training and inference process, whereas GAN only requires one forward propagation to generate an output image. Therefore, it is crucial to explore more efficient architectures and sampling methods that can produce high-quality images while minimizing computational overhead. Additionally, in order to enhance the interpretability and controllability of the model and accurately manipulate specific attributes, objects, or regions in the generated image, it is essential to develop model output interpretation techniques, such as spatial conditions. Lastly, to address issues with noise scheduling and control the sampling process, methods such as learning noise scheduling and continuous time diffusion models can be utilized to maintain the model’s generalization ability while adapting to new tasks.
	Weak interpretability and controllability	
	Difficulty in noise scheduling. To enhance the efficiency of diffusion models, Reference [52] emphasizes the relationship between noise scheduling and optimization strategies, which are key factors influencing the efficiency and performance of diffusion models.	

**Table 6 sensors-25-06672-t006:** Comparison of real-valued and hypercomplex layers in terms of parameters, computation, and expressive capacity.

Layers	Parameter Count	Computational Complexity	Activation	Expressive Capacity
Real-valued dense layers	*n* × *m*	*O* (*n* × *m*)	The activation function is applied independently to the numerical output of each neuron, ensuring that each element is processed individually.	More generic
Hypercomplex layer	(*n* × *m*)/4	The computational complexity increases to approximately *O*(16**n***m*), with a parameter count of 4**n***m* real values. But it obtained rotational invariance.	Apply the activation function to the hypercomplex whole, rather than its individual components.	Enhanced capability to model complex feature distributions and capture multidimensional relationships.

**Table 7 sensors-25-06672-t007:** Performance improvement of different data augmentation methods in object detection tasks (%).

Enhancement Methods		Metrics	Dataset	Performance
Image transformation data augmentation [21]	Baseline	mAP (YOLOv3)	CTSD	mAP0.5: 83.4% mAP0.5:0.95: 53.8%
	Mixup			mAP0.5: 84% mAP0.5:0.95: 55.5%
	Cutout			mAP0.5: 83.9% mAP0.5:0.95: 53.7%
	CutMix			mAP0.5: 85.2% mAP0.5:0.95: 56.4%
	Mosaic			mAP0.5: 85.8% mAP0.5:0.95: 57.6%
	FlexibleCP			mAP0.5: 86.4% mAP0.5:0.95: 61.7%
GAN [29]	Baseline	mAP (YOLOv4)	Original images	mAP0.5: 99.55%
	DCGAN			mAP0.5: 99.07%
	LSGAN			mAP0.5: 99.98%
	WGAN			mAP0.5: 99.45%
Diffusion Model [54]	Baseline	mAP (YOLOX)	PASCAL VOC	52.5%
	Diffusion Model			53.7%
Composite Data Augmentation [7]	Alternative Enhancement Method Based on the Standardization Characteristics of Traffic Signs	mAP (YOLOv5)	TT 100 K (45 categories)	mAP0.5: 85.18% mAP@.5:.95: 65.46%
			TT 100 K (24 categories)	mAP0.5: 88.16% mAP@.5:.95: 66.79%

**Table 8 sensors-25-06672-t008:** Traffic Sign Datasets.

Name	Developer	Total	Categories	Attributes	Description
CTSDB	China	16,164	58	Different time, weather condition, lighting conditions, moving blurring, prohibition signs, indication signs, speed limit signs, etc.	All images in the Chinese Traffic Sign Database are labeled with four corresponding items of symbols and categories. It contains 10,000 detection dataset images and 6164 recognition dataset images [55].
CCTSDB	Zhang Jianming’s team	10,000	3	salt-and-pepper noise, indicator signs, prohibition signs, warning signs	The CSUST Chinese Traffic Sign Detection Benchmark includes original images, resized images, images with added salt-and-pepper noise, and images with adjusted brightness [56].
TT 100 K	Tsinghua University and Tencent	100,000	N/A	different weather, lighting conditions, warning, prohibition, mandatory, uneven categories	The Tsinghua Trent 100 K Tutorial dataset covers 30,000 examples of traffic signs [57].
GTSRB	Germany	50,000	40	class labels	The images of German Traffic Sign Recognition Benchmark have been meticulously annotated with corresponding class labels [58].
LISA Traffic Sign Dataset	America	N/A	47	categories, size, location, occlusion	All images are labeled with categories, size, location, occlusion, and auxiliary road information [59].
Mapillary Traffic Sign Dataset	N/A	100,000	N/A	diverse dataset, street view, over 300 bounding box annotations, different weather, seasons, times of day, urban, rural	It is used for detecting and classifying traffic signs around the world [60].
KUL Belgium Traffic Sign Dataset	N/A	145,000	N/A	resolution of 1628 × 1236 pixels	The dataset was created in 2013 and consists of training and testing sets (2D, 3D) [61].
RTSD	Russia	104,359	156	Recognition task	The Russian Traffic Sign Dataset is mainly used for traffic sign recognition [62,63].
STSD	N/A	10,000	N/A	Different roads, weather, and lighting conditions, high image quality, uniform resolution	The Swedish Traffic Sign Dataset collected data in the local area, suitable for traffic sign recognition and classification tasks [64].

**Table 9 sensors-25-06672-t009:** Traffic Light Datasets.

Name	Developer	Total	Categories	Attributes	Description
LISA Traffic Light Dataset	San Diego, CA, USA	N/A	N/A	Day/night, lighting, weather variation	Images and videos captured under different lighting and weather conditions [65].
Bosch Small Traffic Lights Dataset	Bosch	13,427	Training set: 15 Testing set: 4	Rain, strong light, interference	The training set consists of 5093 images and 10,756 annotated traffic lights; The test set consists of 8334 consecutive images and 13,486 annotated traffic lights [66].
DriveU Traffic Light Dataset	N/A	N/A	N/A	Rich scene attributes, low resolutions, small objects	Additionally, it can accurately annotate objects with small pixels, further enhancing its value for this type of research [67].
LaRA	La Route Automatisée, France	11,179 frames (8 min video)	4 categories (red, green, yellow, blurry)	Resolution 640 × 480	The traffic light video dataset contains four types of annotated labels for traffic light detection [68].
PTL	Shanghai American School Puxi Campus	5000	N/A	Pedestrian + traffic light labels	Pedestrian-Traffic-Lights dataset includes both pedestrians and traffic lights [69].

**Table 10 sensors-25-06672-t010:** Traffic Pedestrian Datasets.

Name	Developer	Year	Total	Attributes	Description
Caltech Pedestrian Detection Benchmark	N/A	2009	250,000 frames, 350,000 bounding boxes, 2300 pedestrians	Resolution 640 × 480, 2300 unique pedestrians	10 h of annotated videos for pedestrian detection [70].
TUD-Brussels Pedestrian	Max Planck Institute for Informatics	2010	1326 annotated images	Resolution 640 × 480, 1326 annotated images of pedestrians, multiple viewing angles	Contains pedestrians mostly captured at small scales and various angles [71].
D^2^-City	University of Southern California, Didi Laboratories	N/A	10,000+ videos (1000 fully annotated, rest partially)	Vehicle, pedestrian, street scene annotations	Large-scale driving recorder dataset with diverse objects and scenarios [72].
CityPersons	Research team from Technical University of Munich	2016	5000+ images (2975 train, 500 val, 1575 test)	Subset of CityScape, person-only annotations	High-quality pedestrian dataset for refined detection tasks [73].
CrowdHuman	Bosch	N/A	15,000 train, 4370 val, 5000 test; 470,000 human instances	Avg. 23 people per image, diverse annotations	Rich pedestrian dataset with strong crowd diversity [74].
NightOwls Dataset	N/A	N/A	N/A	Resolution 1024 × 640, multiple cities & conditions	Focused on nighttime pedestrian detection, includes pedestrians, cyclists, and others [75].
JAAD	York University	2017	346 video clips; 2793 pedestrians	Weather variations, daily driving scenario	Captures joint attention behavior in driving contexts [76].
Elektra (CVC-14)	Universitat Autònoma de Barcelona	2016	3110 train + 2880 test (day); 2198 train + 2883 test (night)	Day/night subsets, 2500 pedestrians	Dataset for day/night pedestrian detection tasks [77].
PANDA	Tsinghua University and Duke University	2020	15,974.6 k bounding boxes, 111.8 k attributes, 12.7 k trajectories, 2.2 k groups	Large-scale fine-grained attributes and groups	Dense pedestrian analysis dataset for pose, attributes, and trajectory research [78].

**Table 11 sensors-25-06672-t011:** Vehicle Datasets.

Name	Developer	Year	Total	Attributes	Description
CCPD	University of Science and Technology of China and Xingtai Financial Holding Group	2018	250,000 car images	License plate position annotations	Chinese City Parking Dataset for license plate detection and recognition [79].
Udacity	Udacity	2016	Dataset 1: 9423 frames (1920 × 1200) Dataset 2: 15,000 frames (1920 × 1200)	Cars, trucks, pedestrians (Dataset 1), Cars, trucks, pedestrians, traffic lights (Dataset 2) Daytime environment	Benchmark dataset for autonomous driving research [80].
ONCE	HUAWEI	2021	1 M LiDAR scenes + 7 M camera images	Cars, buses, trucks, pedestrians, cyclists, Diverse weather environments	One millioN sCenEs dataset for large-scale perception research [81].
NYC3DCars	Cornell University	2013	2000 annotated images 3787 annotated vehicles	Vehicle location, type, geographic location, occlusion degree, time	Dataset of 3D vehicles in geographic context (New York) [82].
CompCars	Chinese University of Hong Kong	2015	Web-nature: 136,726 full-car + 27,618 component images Surveillance: 50,000 front-view images	163 car brands, 1716 vehicle models, Web-nature & surveillance-nature data	Comprehensive Cars dataset for fine-grained vehicle classification and analysis [83].

**Table 12 sensors-25-06672-t012:** Road Datasets.

Name	Developer	Year	Total	Attributes	Description
Highway Workzones	Carnegie Mellon University	2015	N/A	Highway driving, sunny, rainy, cloudy, 6 videos, spring, winter	The dataset can be utilized for training and identifying the boundaries of highway driving areas, as well as detecting changes in driving environments. The images have been accurately labeled with 9 different types of tags [84].
TME Motorway Dataset	Czech Technical University in Prague & University of Parma	2011	28 video clips and 30,000 frames	Highways in northern Italy, different traffic conditions, lighting conditions, two subsets, day, night, resolution of 1024 × 768	Only the vehicles have been annotated [85].
RDD-2020	Indian Institute of Technology, University of Tokyo, and UrbanX Technologies	2021	26,620	road damage	Road Damage Dataset 2020 [86].
CeyMo	University of Moratuwa	2021	N/A	1920 × 1080, various regions, 2887 images (with 4706 instances of road markings across 11 categories)	See More on Roads—A Novel Benchmark Dataset for Road Marking Detection. The test set consists of six distinct scenarios: normal, crowded, glare, night, rain, and shadow [87].
Oxford Road Boundaries	University of Oxford	2021	62,605 labeled samples	Straight roads, parked cars, intersections, different scenarios	Detection task [88].

**Table 13 sensors-25-06672-t013:** Traffic Scene Datasets.

Name	Developer	Year	Total	Attributes	Description
ApolloScape	N/A	N/A	N/A	Different cities, different traffic conditions, high resolution images, RGB video	The dataset is divided into three subsets, which are used for training, validation, and testing. No semantic annotation was provided for testing images. All pixels in the ground truth annotation used for testing images are marked as 255 [89].
BLVD	Xi’an Jiaotong University and Chang’an University	2019	214,900 tracking points, 6004 valid segments	5D Semantics Benchmark, autonomous Driving dataset, low/high density of traffic participants, daytime/nighttime	It contains a total of 4900 objects for 5D intent prediction [90].
BDD100K	BERKELEY ARTIFICIAL INTELLIGENCE RESEARCH	2020	100,000 videos	Diverse Driving Video Database, different geographical, environmental and weather conditions	It includes 10 tasks [91].
SODA10M	Huawei Noah’s Ark Laboratory, Sun Yat-sen University, and Chinese University of Hong Kong	2021	N/A	large-scale 2D dataset	A Large-Scale 2D Self/Semi-Supervised Object Detection Dataset for Autonomous Driving. It contains 10 m unlabeled images and 20 k labeled images with 6 representative object categories [92].
Street Scene	Mitsubishi Electric Research Laboratory and North Carolina State University	2020	46 training video sequences, 35 testing video sequences	Street view, car activities, complex backgrounds, pedestrians, trees, two consecutive summers	The sequences captured from street views [93].
KITTI	Karlsruhe Institute of Technology (KIT) and Toyota Technological Institute at Chicago (TTI-C)	2012	14,999 images	Urban, rural, highway scenes,	It provides 14,999 images and corresponding point clouds for 3D object detection tasks [94].
Cityscapes	A consortium primarily led by Daimler AG	2016	N/A	50 cities, different weather and lighting conditions, 30 categories, roads, pedestrians, vehicles, traffic signs, etc.	Image annotation includes pixel level fine annotation corresponding to the image [95].
nuScenes	Motional (formerly nuTonomy)	2019	1000 scenes, 1400,000 camera images	1200 h, Boston, Pittsburgh, Las Vegas, and Singapore	It provides detailed 3D annotations for 23 object classes and is a key benchmark for 3D detection and tracking [96].
Argoverse	Argo AI, Carnegie Mellon University and Georgia Institute of Technology	2019	prediction dataset of 324,557 scenes	3D tracking dataset	The dataset contains 113 3D annotated scenes and a motion prediction dataset [97].
Waymo Open Dataset	Waymo	2019	1150 scenes	High-resolution sensor data, multiple urban and suburban environments, driving scenarios, day and night, sunny and rainy days	Each scenes with a duration of 20 s [98].

**Table 14 sensors-25-06672-t014:** The mapping relationship of “Enhancement Methods-Improvement Type-Evaluation Indicator”.

Enhancement Methods	Improvement Type	Evaluation Indicator
image transformation	Optimize performance and improve methods for addressing issues such as spatial and lighting invariance, local occlusion, and motion blur in the model.	mAP/mIoU (Mean Intersection over Union)/Accuracy
GAN	By improving cross domain generalization ability, extracting small object detail features, and mining rare scenes, the generalization ability and robustness of the model can be enhanced.	mAP/mIoU/Recall
Diffusion Model	Data generation, multimodal fusion, and domain adaptive optimization in complex scenarios.	mAP/mIoU/Recall
Composite Data Augmentation	Improvements and enhancements have been made in various aspects of small object detection performance, context understanding ability, and multimodal fusion.	mAP/mIoU/Recall

## Data Availability

Partial public dataset download link: China Chinese Traffic Sign Database (CTSDB): https://nlpr.ia.ac.cn/pal/trafficdata/recognition.html (accessed on 1 March 2025); Swedish Traffic Sign Dataset (STSD): https://www.selectdataset.com/dataset/2bf39636f1fbe5cd1ac034c6250c9ade (accessed on 7 March 2025); Udacity: https://github.com/udacity (accessed on 9 March 2025).

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
