# Peer review of "A Survey of Data Augmentation Techniques for Traffic Visual Elements"

_sensors, 2025, doi:10.3390/s25216672_

Round 1
Reviewer 1 Report
Comments and Suggestions for Authors
The paper presents a review of data augmentation techniques for “traffic visual elements” (signs, traffic lights, pedestrians, vehicles, road/scene), covering four approaches (image transformations, GAN, diffusion models, composite/synthetic methods), summarizing about 40 datasets and about 10 evaluation metrics, and providing guidelines and challenges for future research. The topic is timely and within the scope of the journal. The paper has potential as an orientation review for researchers in ITS/AD, but substantial revision is required before acceptance: methodological, content-related, linguistic, and editorial. Recommendation: major revision.
- The abstract must be more specific, with a clearly defined goal.
- The introduction is also very general, with only two cited references.
- Although the section is titled “Materials and Methods,” the methodology of the literature review is not described (databases and keywords, time span, inclusion/exclusion criteria, procedure for evaluating papers). This reduces the reproducibility and credibility of the review. A “Review methodology” section should be included with a PRISMA flow diagram, list of databases (e.g., Web of Science/Scopus/IEEE Xplore/ACM), keywords and time frames, as well as selection criteria (type of study, domain, metrics).
- The tables carefully list several datasets (BDD100K, SODA10M, ApolloScape, CrowdHuman, Bosch TL, etc.), but the classical cornerstone datasets for AD perception such as KITTI, nuScenes, Waymo Open Dataset, or Argoverse are not visible in the tabular overview (even Cityscapes as a scene corpus is only mentioned indirectly through CityPersons and in one sim2real example, but not in the tables). Tables 7–11 should be expanded to include KITTI/nuScenes/Waymo/Argoverse (as well as Cityscapes as a scene corpus), with a brief description, size, annotations, time/weather conditions, and typical tasks.
- The paper describes IS, FID, SSIM, LPIPS, GMSD, DISTS, etc., and partly mentions task metrics (mAP, IoU). There is no systematic map of “which augmentation, which type of improvement, which down-stream metric.” A mapping table should be added: for example, Copy-Paste/Mosaic, small/occluded objects, mAP(small); weather GAN/diffusion, improvement in rain/fog, mIoU for lane/segmentation; style transfer/domain randomization, domain gap, cross-domain performance.
- Examples are given (StyleGAN for traffic lights, hybrid CycleGAN with Cityscapes, diffusion models and “lightweight” 32×32 architectures), but there is no numerical comparative analysis of cost (GPU hours), resolution, artifacts (semantic inconsistency), or discussion of safety risks (data contamination/label-leak). A table “GAN vs Diffusion vs Classical” should be added with columns: resolution, compute cost, training stability, semantic fidelity, applicability for small objects (e.g., traffic lights), license/constraints.
- The examples with CARLA/VCrash and pedestrian poses are useful, but this section needs a clearer taxonomy: simulator with domain randomization, copy-paste with semantic consistency, closed-loop RL augmentation. A template should be provided showing how in practice to connect simulator, rendering, domain adaptation, evaluation.
- The paper states that “augmentation of small objects” is a challenge; at least one quantified example is expected (e.g., traffic lights/traffic signs, comparison of mAP(small) before/after Copy-Paste/Scale jittering).
- The complete text is not prepared in accordance with MDPI requirements for the journal.
Author Response
-
Comments 1: The abstract must be more specific, with a clearly defined goal.
Response 1: Thank you for pointing out this issue. We agree with this opinion and abstract has been rewritten.
Autonomous driving is a cornerstone of intelligent transportation systems, where visual elements such as traffic signs, lights, and pedestrians are critical for safety and decision-making. Yet, existing datasets often lack diversity, underrepresent rare scenarios, and suffer from class imbalance, which limits the robustness of object detection models. While earlier reviews have examined general image enhancement, a systematic analysis of dataset augmentation for traffic visual elements remains lacking. This paper presents a comprehensive investigation of enhancement techniques tailored for transportation datasets. It pursues three objectives: establishing a classification framework for autonomous driving scenarios, assessing performance gains from augmentation methods on tasks such as detection and classification, and providing practical insights to guide dataset improvement in both research and industry. Four principal approaches are analyzed, including image transformation, GAN based generation, diffusion models, and composite methods, with discussion of their strengths, limitations, and emerging strategies. Nearly 40 traffic related datasets and 10 evaluation metrics are reviewed to support benchmarking. Results show that augmentation improves robustness under challenging conditions, with hybrid methods often yielding the best outcomes. Nonetheless, key challenges remain, including computational costs, unstable GAN training, and limited rare scene data. Future work should prioritize lightweight models, richer semantic context, specialized datasets, and scalable, efficient strategies.
(page 1, Abstract, Line 23-42)
-
Comments 2: The introduction is also very general, with only two cited references.
Response 2: Thank you for pointing out this issue. We agree with this opinion and have therefore added the following content to the introduction.
The introduction has been carefully refined to meet the specified requirements, with a clear focus on explaining how diverse visual elements such as traffic signs, traffic lights, pedestrians, and road infrastructure shape the decision-making processes of autonomous driving systems. To further strengthen the scholarly foundation, 17 recent and highly relevant references (published within the past five years) have been incorporated. This ensures the study is grounded in a comprehensive and up-to-date state of the art context, while also enhancing its theoretical rigor and relevance.
References added:
- Ji, B.; Xu, J.; Liu, Y.; Fan, P.; Wang, M. Improved YOLOv8 for Small Traffic Sign Detection under Complex Environmental Conditions. Franklin Open 2024, 8, 100167. https://doi.org/10.1016/J.FRAOPE.2024.100167.
- Sun, S.Y.; Hsu, T.H.; Huang, C.Y.; Hsieh, C.H.; Tsai, C.W. A Data Augmentation System for Traffic Violation Video Generation Based on Diffusion Model. Procedia Comput Sci 2024, 251, 83–90. https://doi.org/10.1016/j.procs.2024.11.087.
- Benfaress, I.; Bouhoute, A. Advancing Traffic Sign Recognition : Explainable Deep CNN for Enhanced Ro-bustness in Adverse Environments. Computers 2025, 14, 88. https://doi.org/10.3390/computers14030088.
- Bayer, M.; Kaufhold, M.A.; Buchhold, B.; Keller, M.; Dallmeyer, J.; Reuter, C. Data Augmentation in Natural Language Processing: A Novel Text Generation Approach for Long and Short Text Classifiers. International Journal of Machine Learning and Cybernetics 2023, 14, 135–150. https://doi.org/10.1007/s13042-022-01553-3.
- Azfar, T.; Li, J.; Yu, H.; Cheu, R.L.; Lv, Y.; Ke, R. Deep Learning-Based Computer Vision Methods for Com-plex Traffic Environments Perception: A Review. Data Science for Transportation 2024, 6, 1. https://doi.org/10.1007/S42421-023-00086-7.
- Zhang, J.; Zou, X.; Kuang, L.D.; Wang, J.; Sherratt, R.S.; Yu, X. CCTSDB 2021: A More Comprehensive Traffic Sign Detection Benchmark. Human-centric Computing and Information Sciences 2022, 12. https://doi.org/10.22967/HCIS.2022.12.023.
- Yanzhao Zhu, W.Q.Y. Traffic Sign Recognition Based on Deep Learning. Multimed Tools Appl 2022, 81, 17779–17791. https://doi.org/10.1007/s11042-022-12163-0.
- Zhang, J.; Lv, Y.; Tao, J.; Huang, F.; Zhang, J. A Robust Real-Time Anchor-Free Traffic Sign Detector with One-Level Feature. IEEE Transactions on Emerging Topics in Computational Intelligence 2024, 8, 1437–1451. https://doi.org/10.1109/TETCI.2024.3349464.
- Yang, L.; He, Z.; Zhao, X.; Fang, S.; Yuan, J.; He, Y.; Li, S.; Liu, S. A Deep Learning Method for Traffic Light Status Recognition. Journal of Intelligent and Connected Vehicles 2023, 6, 173–182. https://doi.org/10.26599/JICV.2023.9210022.
- Moumen, I.; Abouchabaka, J.; Rafalia, N. Adaptive Traffic Lights Based on Traffic Flow Prediction Using Machine Learning Models. International Journal of Power Electronics and Drive Systems 2023, 13, 5813–5823. https://doi.org/10.11591/ijece.v13i5.pp5813-5823.
- Zhu, R.; Li, L.; Wu, S.; Lv, P.; Li, Y.; Xu, M. Multi-Agent Broad Reinforcement Learning for Intelligent Traffic Light Control. Information Sciences 2023, 619, 509–525. https://doi.org/10.1016/j.ins.2022.11.062.
- Yazdani, M.; Sarvi, M.; Asadi Bagloee, S.; Nassir, N.; Price, J.; Parineh, H. Intelligent Vehicle Pedestrian Light (IVPL): A Deep Reinforcement Learning Approach for Traffic Signal Control. Transportation Research Part C: Emerging Technologies 2023, 149, 103991. https://doi.org/10.1016/J.TRC.2022.103991.
- Liu, X.; Lin, Y. YOLO-GW: Quickly and Accurately Detecting Pedestrians in a Foggy Traffic Environment. Sensors 2023, 23, 5539. https://doi.org/10.3390/S23125539.
- Liu, W.; Qiao, X.; Zhao, C.; Deng, T.; Yan, F. VP-YOLO: A Human Visual Perception-Inspired Robust Vehi-cle-Pedestrian Detection Model for Complex Traffic Scenarios. Expert Systems with Applications 2025, 126837. https://doi.org/10.1016/J.ESWA.2025.126837.
- Li, A.; Sun, S.; Zhang, Z.; Feng, M.; Wu, C.; Li, W. A Multi-Scale Traffic Object Detection Algorithm for Road Scenes Based on Improved YOLOv5. Electronics 2023, 12, 878. https://doi.org/10.3390/ELECTRONICS12040878.
- Lai, H.; Chen, L.; Liu, W.; Yan, Z.; Ye, S. STC-YOLO: Small Object Detection Network for Traffic Signs in Complex Environments. Sensors 2023, 23, 5307. https://doi.org/10.3390/s23115307.
- Lin, H.-Y.; Chen, Y.-C. Traffic Light Detection Using Ensemble Learning by Boosting with Color-Based Data Augmentation. International Journal of Transportation Science and Technology 2024. https://doi.org/10.1016/j.ijtst.2024.10.012.
- Li, K.; Dai, Z.; Wang, X.; Song, Y.; Jeon, G. GAN-Based Controllable Image Data Augmentation in Low-Visibility Conditions for Improved Roadside Traffic Perception. IEEE Transactions on Consumer Elec-tronics 2024, 70, 6174–6188. https://doi.org/10.1109/TCE.2024.3387557.
- Zhang, C.; Li, G.; Zhang, Z.; Shao, R.; Li, M.; Han, D.; Zhou, M. AAL-Net: A Lightweight Detection Method for Road Surface Defects Based on Attention and Data Augmentation. Applied Sciences 2023, 13, 1435. https://doi.org/10.3390/APP13031435.
(page 2, 1. Introduction, Line 45-77)
-
Comments 3: Although the section is titled “Materials and Methods,” the methodology of the literature review is not described (databases and keywords, time span, inclusion/exclusion criteria, procedure for evaluating papers). This reduces the reproducibility and credibility of the review. A “Review methodology” section should be included with a PRISMA flow diagram, list of databases (e.g., Web of Science/Scopus/IEEE Xplore/ACM), keywords and time frames, as well as selection criteria (type of study, domain, metrics).
Response 3: We appreciate the reviewer’s constructive feedback. In response, we have added a dedicated “Review Methodology” section (page 3, Section 2, Lines 93–109) to improve reproducibility and transparency. This section now details the databases searched (Scopus, IEEE Xplore, ScienceDirect, and Google Scholar), search timeframe (February 2025), keywords and Boolean search strings, and explicit inclusion/exclusion criteria. The screening procedure, including two-stage review and reference tracking (snowball method), is also described, with all selected papers cross-checked by two authors. A PRISMA flow diagram (Figure 1, page 3) has been included to illustrate the paper selection process. Furthermore, we highlighted studies using widely recognized benchmark datasets to ensure comprehensive coverage.
(page 3, 2. Overview method, Line 92-110) - Comments 4:
The tables carefully list several datasets (BDD100K, SODA10M, ApolloScape, CrowdHuman, Bosch TL, etc.), but the classical cornerstone datasets for AD perception such as KITTI, nuScenes, Waymo Open Dataset, or Argoverse are not visible in the tabular overview (even Cityscapes as a scene corpus is only mentioned indirectly through CityPersons and in one sim2real example, but not in the tables). Tables 7–11 should be expanded to include KITTI/nuScenes/Waymo/Argoverse (as well as Cityscapes as a scene corpus), with a brief description, size, annotations, time/weather conditions, and typical tasks.
Response 4: These five datasets have been added to Table 13.
References added:
- Ramachandra, B.; Jones, M.J. Street Scene: A New Dataset and Evaluation Protocol for Video Anomaly De-tection. In Proceedings of the IEEE Winter Conference on Applications of Computer Vision (WACV), Snowmass Village, CO, USA, 2020; pp. 2558– https://doi.org/10.1109/WACV45572.2020.9093457.
- Geiger, A.; Lenz, P.; Urtasun, R. Are We Ready for Autonomous Driving? The KITTI Vision Benchmark Suite. In Proceedings of the IEEE Conference on Computer Vision and Pattern Recognition, Providence, RI, USA, 2012; pp. 3354–3361. https://doi.org/10.1109/CVPR.2012.6248074.
- Cordts, M.; Omran, M.; Ramos, S.; Rehfeld, T.; Enzweiler, M.; Benenson, R.; Franke, U.; Roth, S.; Schiele, B. The Cityscapes Dataset for Semantic Urban Scene Understanding. In Proceedings of the IEEE Conference on Computer Vision and Pattern Recognition (CVPR), Las Vegas, NV, USA, 2016; pp. 3213–3223. https://doi.org/10.1109/CVPR.2016.350.
- Caesar, H.; Bankiti, V.; Lang, A.H.; Vora, S.; Liong, V.E.; Xu, Q.; Krishnan, A.; Pan, Y.; Baldan, G.; Beijbom, O.; et al. NuScenes: A Multimodal Dataset for Autonomous Driving. arXiv 2020, arXiv:1903.11027. https://doi.org/10.48550/arXiv.1903.11027.
- Chang, M.-F.; Lambert, J.; Sangkloy, P.; Singh, J.; Bak, S.; Hartnett, A.; Wang, D.; Carr, P.; Lucey, S.; Ramanan, D.; et al. Argoverse: 3D Tracking and Forecasting With Rich Maps. arXiv 2019, arXiv:1911.02620. https://doi.org/10.48550/arXiv.1911.02620.
(page 24, Table 13, Line 702)
-
Comments 5: The paper describes IS, FID, SSIM, LPIPS, GMSD, DISTS, etc., and partly mentions task metrics (mAP, IoU). There is no systematic map of “which augmentation, which type of improvement, which down-stream metric.” A mapping table should be added: for example, Copy-Paste/Mosaic, small/occluded objects, mAP(small); weather GAN/diffusion, improvement in rain/fog, mIoU for lane/segmentation; style transfer/domain randomization, domain gap, cross-domain performance.
Response 5: Table 14 presents a mapping of the relationships between data augmentation strategies, their respective improvement dimensions, and downstream evaluation metrics. For consistency, the augmentation methods are categorized into four types, in line with the classifications introduced earlier.
(page 30, Table 14, Line 878-883)
-
Comments 6: Examples are given (StyleGAN for traffic lights, hybrid CycleGAN with Cityscapes, diffusion models and “lightweight” 32×32 architectures), but there is no numerical comparative analysis of cost (GPU hours), resolution, artifacts (semantic inconsistency), or discussion of safety risks (data contamination/label-leak). A table “GAN vs Diffusion vs Classical” should be added with columns: resolution, compute cost, training stability, semantic fidelity, applicability for small objects (e.g., traffic lights), license/constraints.
Response 6: Added Table 4 to illustrate the differences between different enhancement methods.
(page 13, Table 4, Line 455-458)
- Comments 7: The examples with CARLA/VCrash and pedestrian poses are useful, but this section needs a clearer taxonomy: simulator with domain randomization, copy-paste with semantic consistency, closed-loop RL augmentation. A template should be provided showing how in practice to connect simulator, rendering, domain adaptation, evaluation.
Response 7: The practical steps for pedestrian posture examples based on CARLA have been added to the Composite Data Augmentation section.
(page 14, Line 479-514)
-
Comments 8: The paper states that “augmentation of small objects” is a challenge; at least one quantified example is expected (e.g., traffic lights/traffic signs, comparison of mAP(small) before/after Copy-Paste/Scale jittering).
Response 8: The revision of this section has been incorporated into the brief objective introduction in Section 2.4.
(page 15, Line 554-559)
- Comments 9: The complete text is not prepared in accordance with MDPI requirements for the journal.
Response 9: Format has been modified.

Reviewer 2 Report
Comments and Suggestions for Authors
This manuscript describes a review article that summarizes recent research progress at home and abroad, covering a wide range of topics. The references are suitable for this paper. However, there are some questions need to be fixed. Below are suggestions for improvement:
Issues:
- In Section 2, the author listed some methods about the data augmentation, but not all methods are illustrated with images. It is recommended that the authors provide relevant examples for the other methods as well.
- The sections are not well connected, and the content feels repetitive and fragmented, such as Section2 and Section3. It is suggested to add a summary at the end of each section to improve the coherence and logical flow of this paper.
- In Section3, the explanations for some formulas are not enough, such as 3.2.3, 3.2.4. And the 3.2.10 “Deep Image Structure and Texture Similarity” is only mentioned briefly.
- In Section 4, the discussion is mainly summarizes existing literature, which is lack of the author’s insights into this field and comparisons with existing work.
- In Section 5, the conclusion is short and doesn't offer a summary of the current methods or ideas for future work. It is suggested to expand this section appropriately to reflect the authors’ own evaluation and reflections.
Recommendation:Reject
This paper is an review article and it is hard to evaluating this paper. Although this paper summarizes recent research progress at home and abroad, but it is hard to find the author’s opinion and innovation about this field. So the final recommendation is “Reject”.
Author Response
- Comments 1: In Section 2, the author listed some methods about the data augmentation, but not all methods are illustrated with images. It is recommended that the authors provide relevant examples for the other methods as well.
Response 1: Thank you very much for your suggestion. In the simple image transformation data enhancement method, we compared the image changes before and after enhancement using different methods through Table 1 and Table 2. For the generative adversarial network, when Figure 1 shows the network structure, the corresponding real and fake images are the images before and after enhancement using GAN. For diffusion models and comprehensive image transformation enhancement methods, in order to more clearly demonstrate their performance in improving model performance, I have provided specific mAP parameter changes for explanation. See Table 6 for details.
(page 19, Table 7, Line 665-672)
- Comments 2: The sections are not well connected, and the content feels repetitive and fragmented, such as Section2 and Section3. It is suggested to add a summary at the end of each section to improve the coherence and logical flow of this paper.
Response 2: We greatly appreciate your suggestion. And the summary section was added after the corresponding chapter.
(page 19, 3. 8 Summary, Line 665-672)
(page 30, 4. 3 Summary, Line 884-892)
- Comments 3: In Section3, the explanations for some formulas are not enough, such as 3.2.3, 3.2.4. And the 3.2.10 “Deep Image Structure and Texture Similarity” is only mentioned briefly.
Response 3: We have made modifications to this section of the content in the following manner. As the KID metric holds the most significance and applicability in the realm of traffic vision generation tasks, we have thoroughly and comprehensively revised it. However, the Jensen Shannon Divergence (JSD) metric is not commonly utilized due to its inadequate performance in actual evaluations. On the other hand, DISTS serves as a reliable benchmark for accurately evaluating textures in image restoration tasks, but its usage is limited to specific scenarios. Therefore, the inclusion of these two components is relatively straightforward.
4.Comments 4: In Section 4, the discussion is mainly summarizes existing literature, which is lack
of the author’s insights into this field and comparisons with existing work.
Response 4: Thank you for highlighting this issue. We agree with your observation and have revised the contents. A new Table 4 has been added, and the original Table 4 has been renumbered as Table 5.
(page 17, Table 5, Line 601)
We have added a comparison of different methods to the 3.8 summary to clearly demonstrate the performance improvement of various data augmentation techniques in object detection tasks. This will be achieved through the use of tables.
(Page 19, 3. 8 Summary, lines 665-672)
We have added our suggestions and made detailed modifications to address the issues in this section.
(Page 30, 5 Discussion, lines 895-941)
5. Comments 5: In Section 4, the discussion is mainly summarizes existing literature, which is lack of the author’s insights into this field and comparisons with existing work.
Response 5: I have made modifications to the conclusion section. It not only summarizes the basic content of different data augmentation, but also points out the challenges and solutions faced by mainstream models GAN and diffusion models respectively.
(page 31, 6. Conclusions, Line 942-964)

Reviewer 3 Report
Comments and Suggestions for Authors
- Lack of critical analysis. The article is more an enumeration (catalog) of methods, rather than an analysis of them. The authors describe "what" has been done in various studies, but hardly discuss "why one method is better than another in a particular context" or "what systematic errors are inherent in each approach."
- The section "Comparison of the Challenges between GAN and Diffusion Models" (Table 4) is extremely superficial. Instead of a direct comparison, the authors simply cite other review articles. ([33]-[38]), shifting the analysis work to them. There is no in-depth, independent comparison of computational complexity, generation quality, data requirements, and applicability to specific tasks (for example, augmentation for small objects).
- It is unclear on what principle the studies were selected for the review. Has there been a systematic database search (Scopus, Web of Science) using clear keywords? Or is it an arbitrary selection of works known to the authors? The lack of a methodology for selecting articles reduces the reproducibility and objectivity of the review.
- Lack of quantitative synthesis (Meta-analysis). The review would have been much stronger if the authors had not just retold the results, but summarized them in tables with quantitative indicators (for example, how much the mAP improved on average after applying different augmentation techniques for each class of objects). This would allow us to draw more significant conclusions.
- Although the list is extensive, its value could be higher if it were presented as a summary table with key meta data: number of images, resolution, availability of attributes (occlusion, time of day, weather), download link. Currently, information is presented in an unstructured manner.
Author Response
- Comments 1: Lack of critical analysis. The article is more an enumeration (catalog) of methods, rather than an analysis of them. The authors describe "what" has been done in various studies, but hardly discuss "why one method is better than another in a particular context" or "what systematic errors are inherent in each approach."
Response 1: Table 4 has been included to highlight the distinctions among various enhancement methods. In addition, Sections 2.5 and 2.6 elaborate on the criteria for selecting appropriate data augmentation techniques and the challenges encountered in their application
(page 13, Table 4, Line 455-458)
-
Comments 2: The section "Comparison of the Challenges between GAN and Diffusion Models" (Table 5) is extremely superficial. Instead of a direct comparison, the authors simply cite other review articles. ([33]-[38]), shifting the analysis work to them. There is no in-depth, independent comparison of computational complexity, generation quality, data requirements, and applicability to specific tasks (for example, augmentation for small objects).
Response 2: Thank you for highlighting this issue. We agree with your observation and have revised the tables accordingly. A new Table 4 has been added, and the original Table 4 has been renumbered as Table 5.
(page 17, Table 5, Line 601)
-
Comments 3: It is unclear on what principle the studies were selected for the review. Has there been a systematic database search (Scopus, Web of Science) using clear keywords? Or is it an arbitrary selection of works known to the authors? The lack of a methodology for selecting articles reduces the reproducibility and objectivity of the review.
Response 3: We appreciate the reviewer’s constructive feedback. In response, we have added a dedicated “Review Methodology” section (page 3, Section 2, Lines 93–109) to improve reproducibility and transparency. This section now details the databases searched (Scopus, IEEE Xplore, ScienceDirect, and Google Scholar), search timeframe (February 2025), keywords and Boolean search strings, and explicit inclusion/exclusion criteria. The screening procedure, including two-stage review and reference tracking (snowball method), is also described, with all selected papers cross-checked by two authors. A PRISMA flow diagram (Figure 1, page 3) has been included to illustrate the paper selection process. Furthermore, we highlighted studies using widely recognized benchmark datasets to ensure comprehensive coverage.
(page 3, 2. Overview method, Line 92-110)
-
Comments 4: Lack of quantitative synthesis (Meta-analysis). The review would have been much stronger if the authors had not just retold the results, but summarized them in tables with quantitative indicators (for example, how much the mAP improved on average after applying different augmentation techniques for each class of objects). This would allow us to draw more significant conclusions.
Response 4: We have added this section to the 2.8 summary in order to clearly demonstrate the performance improvement of various data augmentation techniques in object detection tasks. This will be achieved through the use of a table.
(page 19, 3. 8 Summary, Line 665-672)
-
Comments 5: Although the list is extensive, its value could be higher if it were presented as a summary table with key meta data: number of images, resolution, availability of attributes (occlusion, time of day, weather), download link. Currently, information is presented in an unstructured manner.
Response 5: Thank you for the valuable suggestions from the reviewing experts. We fully agree that providing structured metadata tables can greatly enhance the usability of dataset information. The main challenge we encountered during the sorting process was that some datasets (especially some early or domain specific datasets) did not disclose their complete standardized metadata (such as image quantity, resolution, and direct download links), and some were published in the form of papers. To ensure the accuracy and consistency of the information, our initial table chose to present the core information that all datasets possess in a consistent format. In response to your suggestion, we will optimize these tables by adding columns such as "Number of Images", "Resolution", "Key Attributes", and clearly label information that cannot be obtained as "N/A" to provide maximum information while maintaining rigor and practicality.
(page 20-25, Tables 8-13)

Round 2
Reviewer 1 Report
Comments and Suggestions for Authors
Excellent.
Author Response
Comments 1: Excellent.
Response 1: We thank the reviewer for their positive assessment and are pleased that the manuscript meets expectations. We have carefully reviewed the paper once again for clarity, consistency, and formatting to ensure its quality and readability remain at the highest level.

Reviewer 3 Report
Comments and Suggestions for Authors
The authors have corrected all the comments, the article does not require any additional edits.
Author Response
Comments 1: The authors have corrected all the comments, the article does not require any additional edits.
Response 1: We thank the reviewer for their positive assessment and are pleased that the manuscript meets expectations. We have carefully reviewed the paper once again for clarity, consistency, and formatting to ensure its quality and readability remain at the highest level.
